# ADVERSARIALLY ROBUST AND PRIVACY-PRESERVING REPRESENTATION LEARNING VIA INFORMATION THEORY

## ABSTRACT

Machine learning models are vulnerable to both security attacks (e.g., adversarial examples) and privacy attacks (e.g., private attribute inference). Existing defenses propose different strategies to individually defend against the security attack or privacy attack, and combining them would yield suboptimal performance. In this paper, we aim to mitigate both the security and privacy attacks, and maintain utility of the learning task simultaneously. We achieve the goal by proposing a representation learning framework based on information theory, i.e., learning information-theoretic representations that are robust to adversarial examples and attribute inference adversaries, and effective for learning tasks as well. We also derive novel theoretical results, e.g., the inherent tradeoff between adversarial robustness/utility and attribute privacy, and guaranteed attribute privacy leakage against attribute inference adversaries.

## 1 INTRODUCTION

Machine learning (ML) has achieved remarkable breakthroughs in many research fields, including but not limited to computer vision, speech, and natural language processing. However, recent works show that the current ML design is vulnerable to both security and privacy attacks, e.g., adversarial examples and private attribute inference. Adversarial examples (Szegedy et al., 2013; Goodfellow et al., 2015; Carlini & Wagner, 2017) are typically generated by carefully adding imperceptible perturbations to natural data and they remain a serious problem that prevents the deployment of modern ML models in safety-critical applications such as autonomous driving (Eykholt et al., 2018) and medical imaging (Bortsova et al., 2021). In addition, many real-world applications involve data that contain sensitive/private information, such as race, gender, income, and age. When applying ML to these applications, it poses a great challenge since private attribute can often be accurately inferred (Jia et al., 2017; Aono et al., 2017; Melis et al., 2019).

To mitigate adversarial examples and attribute inference attacks, many defenses have been proposed, but they mainly follow two separate lines and with different techniques. For instance, the state-of-the-art defenses against adversarial examples are based on adversarial training (Madry et al., 2018; Zhang et al., 2019; Wang et al., 2019), which solves a min-max optimization problem. In contrast, the representative defense against inference attacks are based on differential privacy (Abadi et al., 2016), which is a statistical method (more details on defenses against adversarial examples and attribute inference attacks are in Section 2). Some works (Song et al., 2019b;a) show that adversarially robust models only can even leak more private information (also verified in our Section 5.2). In addition, we observe that combining the state-of-the-art defenses against adversarial examples and attribute inference attacks produce suboptimal performance (see results in Section 5.3).

In this paper, we focus on the research question: *1) Can we design an adversarially robust and attribute privacy protection model, while maintaining utility of (unknown) downstream tasks simultaneously? 2) Further, can we theoretically understand the relationships among adversarial robustness, utility, and attribute privacy?* To achieve the goal, we propose an information-theoretic defense framework through the lens of *representation learning*, termed **ARPRL**. Representation learning is very pertinent in today's context given the rise of foundation/large ML models. Particularly, instead of training large models from scratch, which requires huge computational resources and is time consuming, shared learnt representations ensures the community to save much time and costs. Our ARPRL is partly inspired by Zhu et al. (2020); Zhou et al. (2022), which show adversarially robust representations based defenses outperform the *de facto* adversarial training based methods, while being *the first work* to non-trivially generalize learning data representations that are robust to both adversarial examples and attribute inference adversaries. More specifically, we formulate learning representations via three mutual information (MI) objectives: one for adversarial robustness, one for attribute privacy protection, and one for utility preservation. We point out that our ARPRL is *task-agnostic*, meaning the learnt representations does not need to know the target task at hand and can be used for any downstream task. However, directly solving the MI objectives is challenging, as calculating an MI between two arbitrary variables is often infeasible (Peng et al., 2019). To address it, we are motivated by the MI neural estimation (Alemi et al., 2017; Belghazi et al., 2018;

Oord et al., 2018; Poole et al., 2019; Hjelm et al., 2019; Cheng et al., 2020), which converts the intractable MI calculations to the tractable variational MI bounds. Then we parameterize each bound with a neural network, and finally train the neural networks to approximate the true MI.

Based on our designed MI objectives, we can derive novel theoretical results. For instance, we obtain an inherent tradeoff between adversarial robustness and attribute privacy, as well as between utility and attribute privacy. These tradeoffs are also verified through the experimental evaluations on multiple benchmark datasets. We also derive the guaranteed attribute privacy leakage against (worst-case) attribute inference adversaries. Our key contributions can be summarized below:

- This is the first work to advocate learning both robust and privacy-preserving ML models from the representation learning perspective.
- We formulate learning adversarially robust and privacy-preserving representations via information theory—an elegant yet powerful tool.
- Under the information-theoretic framework, we derive novel theoretical results: the tradeoff among adversarial robustness, utility, and attribute privacy, and guaranteed attribute privacy leakage.

## 2 RELATED WORK

**Defenses against adversarial examples.** Many efforts have been made to improve the adversarial robustness of ML models against adversarial examples (Goodfellow et al., 2015; Kurakin et al., 2017; Pang et al., 2019; Wong & Kolter, 2018; Mao et al., 2019; Cohen et al., 2019; Zhai et al., 2020; Wong et al., 2020). Among them, adversarial training based defenses (Madry et al., 2018; Zhang et al., 2019; Wang et al., 2019; Dong et al., 2020; Zhou et al., 2021) has become the mainstream defense and achieved the state-of-the-art defense effectiveness. At a high level, adversarial training augments training data with adversarial examples (e.g., via FGSM attack (Szegedy et al., 2013), CW attack (Carlini & Wagner, 2017), PGD attack (Madry et al., 2018), AutoAttack (Croce & Hein, 2020)) and uses a min-max formulation to train the target ML model (Madry et al., 2018). However, as pointed out by Zhu et al. (2020); Zhou et al. (2022), the dependence between the output of the target model and the input/adversarial examples has not been well studied, which makes the ability of adversarial training not fully exploited. To improve it, Zhu et al. (2020); Zhou et al. (2022) propose to learn adversarially-robust representations via mutual information, which is shown to outperform the state-of-the-art adversarial training based defenses. Our ARPRL is inspired by them while having a nontrivial generalization to learn both robust and privacy-preserving representations.

**Defenses against inference attacks.** Existing privacy-preserving methods against inference attacks can be roughly classified as *adversarial learning* (Oh et al., 2017; Wu et al., 2018; Pittaluga et al., 2019; Liu et al., 2019), *differential privacy* (Shokri & Shmatikov, 2015; Abadi et al., 2016), and *information obfuscation* (Bertran et al., 2019; Hamm, 2017; Osia et al., 2020b; Roy & Boddeti, 2019; Zhao et al., 2020; Osia et al., 2020a; Li et al., 2021). Adversarial learning methods are mainly inspired by GAN (Goodfellow et al., 2014) and they learn obfuscated features from the training data so that their privacy information cannot be inferred from a learnt model. However, these methods need to know the primary task in advance and lack of formal privacy guarantees. Differential privacy methods have formal privacy guarantees, but they have high utility losses. Information obfuscation methods aim to maximize the utility, under the constraint of bounding the information leakage, but almost all of them are empirical and task-dependent. The only exception is Zhao et al. (2020), which has guaranteed information leakage. However, this works requires stronger assumptions (e.g., conditional independence assumption between variables). Our work can be seen as a combination of information obfuscation with adversarial learning to learn both robust and privacy-preserving representations. It provides privacy leakage guarantees as well as inherent tradeoffs between robustness/utility and privacy.

## 3 PRELIMINARIES AND PROBLEM SETUP

**Notations.** We use $s$, $\mathbf{s}$, and $\mathcal{S}$ to denote (random) scalar, vector, and space, respectively. Given a data $\mathbf{x} \in \mathcal{X}$, we denote its label as $y \in \mathcal{Y}$ and private attribute as $u \in \mathcal{U}$, where $\mathcal{X}$, $\mathcal{Y}$, and $\mathcal{U}$ are input data space, label space, and attribute space, respectively. An $l_p$ ball centered at a data $\mathbf{x}$ with radius $\epsilon$ is defined as $\mathcal{B}_p(\mathbf{x}, \epsilon) = \{\mathbf{x}' \in \mathcal{X} : \|\mathbf{x}' - \mathbf{x}\|_p \leq \epsilon\}$. The joint distribution of $\mathbf{x}$, $y$, and $u$ is denoted as $\mathcal{D}$. We further denote $f : \mathcal{X} \to \mathcal{Z}$ as the representation learner that maps $\mathbf{x} \in \mathcal{X}$ to its representation $\mathbf{z} \in \mathcal{Z}$, where $\mathcal{Z}$ is the representation space. Moreover, we let $C : \mathcal{Z} \to \mathcal{Y}$ be the *primary task classifier*, which predicts data label $y$ based on the learnt data representation $\mathbf{z}$, and $A : \mathcal{Z} \to \mathcal{U}$ be the *attribute inference classifier*, which infers the private attribute $u$ based on the representation $\mathbf{z}$. The composition function of two functions $f$ and $g$ is denoted as $(g \circ f)(\mathbf{x}) = g(f(\mathbf{x}))$. We use $[m]$ to denote the set $\{1, 2, \cdots, m\}$ and $|\cdot|$ to denote its cardinality.

**Mutual information (MI) and entropy.** In information theory, MI is a measure of shared information between two random variables, and offers a quantifiable metric for the amount of information leakage on one variable

given the other. Let $(\mathbf{x}, \mathbf{z})$ be a pair of random variables with values over the space $\mathcal{X} \times \mathcal{Z}$. Then the MI of $\mathbf{x}$ and $\mathbf{z}$ is defined as

$$I(\mathbf{x}; \mathbf{z}) = \int_{\mathcal{Z}} \int_{\mathcal{X}} p(\mathbf{x}, \mathbf{z}) \log \frac{p(\mathbf{x}, \mathbf{z})}{p(\mathbf{x})p(\mathbf{z})} d\mathbf{x} d\mathbf{z}. \tag{1}$$

Intuitively, $I(\mathbf{x}, \mathbf{z})$ tells us how well one can predict $\mathbf{z}$ from $\mathbf{x}$ (and $\mathbf{x}$ from $\mathbf{z}$, since it is symmetric). By definition, $I(\mathbf{x}, \mathbf{z}) = 0$ if $\mathbf{x}$ and $\mathbf{z}$ are independent, i.e., $\mathbf{x} \perp \mathbf{z}$. On the other hand, when $\mathbf{x}$ and $\mathbf{z}$ are identical, $I(\mathbf{x}; \mathbf{x}) = H(\mathbf{x}) = \int_{\mathcal{X}} -p(\mathbf{x}) \log p(\mathbf{x}) d\mathbf{x}$, which is the entropy of $\mathbf{x}$.

**Adversarial example/perturbation, adversarial risk, and representation vulnerability Zhu et al. (2020).** Let $\mathcal{X}$ and $\mathcal{Y}$ be the data space and label space, respectively, and $\epsilon$ as the $l_p$ perturbation budget. For any classifier $C : \mathcal{X} \to \mathcal{Y}$, the *adversarial risk* of $C$ with respect to $\epsilon$ is defined as:

$$\text{AdvRisk}_\epsilon(C) = \Pr[\exists \mathbf{x}' \in \mathcal{B}_p(\mathbf{x}, \epsilon), \text{ s.t. } C(\mathbf{x}') \neq y] = \sup_{\mathbf{x}' \in \mathcal{B}_p(\mathbf{x}, \epsilon)} \Pr[C(\mathbf{x}') \neq y], \tag{2}$$

where $\mathbf{x}'$ is called *adversarial example* and $\delta = \mathbf{x}' - \mathbf{x}$ is *adversarial perturbation* with $\|\delta\|_p \leq \epsilon$. Formally, adversarial risk captures the vulnerability of a classifier to adversarial perturbations. When $\epsilon = 0$, adversarial risk reduces to the standard risk, i.e., $\text{AdvRisk}_0(C) = \text{Risk}(C) = \Pr(C(\mathbf{x}) \neq y)$.

Motivated by the empirical and theoretical difficulties of robust learning with adversarial examples, Zhu et al. (2020); Zhou et al. (2022) target learning adversarially robust representations based on MI. They introduced the term *representation vulnerability*: Given a representation learner $f : \mathcal{X} \to \mathcal{Z}$ and an $l_p$ perturbation budget $\epsilon$, the representation vulnerability of $f$ with respect to $\epsilon$ is defined as

$$\text{RV}_\epsilon(f) = \max_{\mathbf{x}' \in \mathcal{B}_p(\mathbf{x}, \epsilon)} [I(\mathbf{x}; \mathbf{z}) - I(\mathbf{x}'; \mathbf{z}')], \tag{3}$$

where $\mathbf{z} = f(\mathbf{x})$ and $\mathbf{z}' = f(\mathbf{x}')$ are the learnt representation for $\mathbf{x}$ and $\mathbf{x}'$, respectively. We note that *higher/smaller* $\text{RV}_\epsilon(f)$ *values imply the representation is less/more robust to adversarial perturbations*. Further, Zhu et al. (2020) linked the connection between adversarial robustness and representation vulnerability through the following theorem:

**Theorem 1** (Zhu et al. (2020)). *Consider all the primary task classifiers as* $\mathcal{C} = \{C : \mathcal{Z} \to \mathcal{Y}\}$. *Given the perturbation budget* $\epsilon$, *for any representation learner* $f : \mathcal{X} \to \mathcal{Z}$,

$$\inf_{C \in \mathcal{C}} \text{AdvRisk}_\epsilon(C \circ f) \geq 1 - \big(I(\mathbf{x}; \mathbf{z}) - \text{RV}_\epsilon(f) + \log 2\big) / \log |\mathcal{Y}|. \tag{4}$$

The theorem states that a smaller representation vulnerability implies a smaller adversarial risk, which means better adversarial robustness, and vice versa. Finally, $f$ is called $(\epsilon, \tau)$-robust if $\text{RV}_\epsilon(f) \leq \tau$.

**Attribute inference attacks and advantage.** Without loss of generality, we assume the attribute space $\mathcal{U}$ is binary. Let $\mathcal{A}$ be the set of all binary attribute inference classifiers that takes data representations $\mathbf{z} = f(\mathbf{x})$ as an input and infers the private attribute $u$, i.e., $\mathcal{A} = \{A : \mathcal{Z} \to \mathcal{U} = \{0, 1\}\}$. Then, we formally define the *attribute inference advantage* of the worst-case attribute inference adversary with respect to the joint distribution $\mathcal{D} = \{\mathbf{x}, y, u\}$ as below:

$$\text{Adv}_\mathcal{D}(\mathcal{A}) = \max_{A \in \mathcal{A}} |\Pr_\mathcal{D}(A(\mathbf{z}) = a | u = a) - \Pr_\mathcal{D}(A(\mathbf{z}) = a | u = 1 - a)|, \forall a = \{0, 1\}. \tag{5}$$

We can observe that: if $\text{Adv}_\mathcal{D}(\mathcal{A}) = 1$, an adversary can *completely* infer the privacy attribute through the learnt representations. In contrast, if $\text{Adv}_\mathcal{D}(\mathcal{A}) = 0$, an adversary obtains a *random guessing* inference performance. To protect the private attribute, we aim to obtain a small $\text{Adv}_\mathcal{D}$.

**Threat model and problem setup.** We focus on a classification task under the adversarial setting. We consider the attacker's goal is to perform both attribute inference and adversarial example attacks. *We assume the attacker does not have access to the internal representation learner (i.e., $f$), but instead can obtain and arbitrarily use the shared data representations.* [1] The attacker is also assumed to have some background knowledge (e.g., even know the underlying data distribution). As the defense is task-agnostic, the defender does not know the learning task. Our goal is to learn task-agnostic representations that are adversarially robust, protect attribute privacy, and maintain the utility of (unknown) downstream tasks. Formally, given $\{\mathbf{x}, y, u\}$ from an underlying distribution $\mathcal{D}$, and a perturbation budget $\epsilon$, we aim to obtain the representation learner $f$ such that the representation vulnerability $RV_\epsilon(f)$ is small, attribute inference advantage $\text{Adv}_\mathcal{D}(\mathcal{A})$ is small, and the risk $\text{Risk}(C)$ is small.

---

[1]This is practical when representation learner is deployed as an API: end-users obtain the representations via querying the API with their data, but do not know the details about the representation learner. Note that many companies have deployed representation learner as an API to provide the machine learning service, e.g., Amazon's AWS Marketplace (AWS Marketplace), OpenAI's Embedding API (cha), and Clarifai's General Embedding API (Clarifai).

## 4 DESIGN OF ARPRL

In this section, we will design our **a**dversarilly **r**obust and **p**rivacy-preserving **r**epresentation **l**earning method, termed **ARPRL**, inspired by information theory.

### 4.1 FORMULATING ARPRL VIA MI OBJECTIVES

Given a data $\mathbf{x}$ with private attribute $u$ sampled from a distribution $\mathcal{D}$, and a perturbation budget $\epsilon$, our purpose is to convert $\mathbf{x}$ into a representation $\mathbf{z} = f(\mathbf{x})$ that satisfies the following three goals:

- **Goal 1: Privacy protection.** $\mathbf{z}$ contains as less information as possible about the private attribute $u$. Ideally, when $\mathbf{z}$ does not include information about $u$, i.e., $\mathbf{z} \perp u$, it is impossible to infer $u$ from $\mathbf{z}$.

- **Goal 2: Utility preservation.** $\mathbf{z}$ should be be useful for many downstream tasks. To achieve the goal, we require $\mathbf{z}$ should include as much information about the data $\mathbf{x}$ as possible, while excluding the private attribute $u$. Ideally, when $\mathbf{z}$ retains the most information about $\mathbf{x}$, the model trained on $\mathbf{z}$ will have the same performance as the model trained on the raw $\mathbf{x}$ (though we do not know the downstream task), thus preserving utility.

- **Goal 3: Adversarially robust.** $\mathbf{z}$ should be not sensitive to adversarial perturbations on the data $\mathbf{x}$, indicating a small representation vulnerability.

We propose to formalize the above goals via MI. Formally, we quantify the goals as below:

$$\textbf{Formalizing Goal 1:} \quad \min_f I(\mathbf{z}; u); \tag{6}$$

$$\textbf{Formalizing Goal 2:} \quad \max_f I(\mathbf{x}; \mathbf{z}|u); \tag{7}$$

$$\textbf{Formalizing Goal 3:} \quad \min_f \left\{ RV_\epsilon(f|u) = \max_{\mathbf{x}' \in \mathcal{B}_p(\mathbf{x},\epsilon)} [I(\mathbf{x}; \mathbf{z}|u) - I(\mathbf{x}'; \mathbf{z}'|u)] \right\}. \tag{8}$$

where 1) we minimize $I(\mathbf{z}; u)$ to maximally reduce the correlation between $\mathbf{z}$ and the private attribute $u$; 2) $I(\mathbf{x}; \mathbf{z}|u)$ is the MI between $\mathbf{x}$ and $\mathbf{z}$ given $u$. We maximize such MI to keep the raw information in $\mathbf{x}$ as much as possible in $\mathbf{z}$ and remove the information that $\mathbf{x}$ contains about the private $u$; 3) $RV_\epsilon(f|u)$ is the representation vulnerability of $f$ conditional on $u$ with respect to $\epsilon$. Minimizing it learns adversarially robust representations that exclude the information about private $u$. Note that $I(\mathbf{x}; \mathbf{z}|u)$ in Equation (8) can be merged with that in Equation (7). Hence Equation (8) can be reduced to the below min-max optimization problem:

$$\max_f \min_{\mathbf{x}' \in \mathcal{B}_p(\mathbf{x},\epsilon)} I(\mathbf{x}'; \mathbf{z}'|u). \tag{9}$$

**Objective function of ARPRL:** Combining the above equations, we have the MI objective function to learn adversarially robust and privacy preserving representations as below:

$$\max_f \left[ -\alpha I(\mathbf{z}; u) + \beta \min_{\mathbf{x}' \in \mathcal{B}_p(\mathbf{x},\epsilon)} I(\mathbf{x}'; \mathbf{z}'|u) + (1 - \alpha - \beta)I(\mathbf{x}; \mathbf{z}|u) \right], \tag{10}$$

where $\alpha, \beta \in [0, 1]$ are tradeoff hyperparameters. That is, a larger/smaller $\alpha$ indicates a stronger/weaker attribute privacy protection and a larger/smaller $\beta$ indicates a stronger/weaker robustness against adversarial perturbations.

### 4.2 ESTIMATING MI VIA TRACTABLE VARIATIONAL BOUNDS

The key challenge of solving Equation (10) is that calculating an MI between two arbitrary random variables is likely to be infeasible (Peng et al., 2019). To address it, we are inspired by the existing MI neural estimation methods (Alemi et al., 2017; Belghazi et al., 2018; Oord et al., 2018; Poole et al., 2019; Hjelm et al., 2019; Cheng et al., 2020), which convert the intractable exact MI calculations to the tractable variational MI bounds. Then, we parameterize each variational MI bound with a neural network, and train the neural networks to approximate the true MI. *We clarify that we do not design novel MI neural estimators, but adopt existing ones to assist our customized MI terms for learning adversarially robust and privacy-preserving representations.*

**Minimizing upper bound MI in Equation (6) for privacy protection.** We propose to adapt the variational upper bound CLUB proposed in (Cheng et al., 2020). Specifically,

$$I(\mathbf{z}; u) \leq I_{vCLUB}(\mathbf{z}; u) = \mathbb{E}_{p(\mathbf{z},u)}[\log q_\Psi(u|\mathbf{z})] - \mathbb{E}_{p(\mathbf{z})p(u)}[\log q_\Psi(u|\mathbf{z})], \tag{11}$$

where $q_\Psi(u|\mathbf{z})$ is an auxiliary posterior distribution of $p(u|\mathbf{z})$ and it needs to satisfy the condition: $KL(p(\mathbf{z}, u)||q_\Psi(\mathbf{z}, u)) \leq KL(p(\mathbf{z})p(u)||q_\Psi(\mathbf{z}, u))$. To achieve this, we need to minimize:

$$\min_\Psi KL(p(\mathbf{z}, u)||q_\Psi(\mathbf{z}, u)) = \min_\Psi KL(p(u|\mathbf{z})||q_\Psi(u|\mathbf{z}))$$

$$= \min_\Psi \mathbb{E}_{p(\mathbf{z},u)}[\log p(u|\mathbf{z})] - \mathbb{E}_{p(\mathbf{z},u)}[\log q_\Psi(u|\mathbf{z}))] \iff \max_\Psi \mathbb{E}_{p(\mathbf{z},u)}[\log q_\Psi(u|\mathbf{z})], \tag{12}$$

where we use that $\mathbb{E}_{p(\mathbf{z},u)}[\log p(u|\mathbf{z})]$ is irrelevant to $\Psi$.

Finally, our **Goal 1** for privacy protection is reformulated as solving the min-max objective function:

$$\min_f \min_\Psi I_{vCLUB}(\mathbf{z};u) \iff \min_f \max_\Psi \mathbb{E}_{p(\mathbf{z},u)}[\log q_\Psi(u|\mathbf{z})]. \tag{13}$$

*Remark.* We note that Equation (13) can be interpreted as an *adversarial game* between: (1) an adversary $q_\Psi$ (i.e., attribute inference classifier) who aims to infer the private attribute $u$ from the representation $\mathbf{z}$; and (2) a defender (i.e., the representation learner $f$) who aims to protect the private attribute $u$ from being inferred.

**Maximizing lower bound MI in Equation (7) for utility preservation.** We adopt the MI estimator proposed in Nowozin et al. (2016) to estimate the lower bound of the MI Equation (7). Specifically,

$$\begin{aligned}
I(\mathbf{x};\mathbf{z}|u) &= H(\mathbf{x}) - H(\mathbf{x}|\mathbf{z},u) \\
&= H(\mathbf{x}) + \mathbb{E}_{p(\mathbf{x},\mathbf{z},u)}[\log p(\mathbf{x}|\mathbf{z},u)] \\
&= H(\mathbf{x}) + \mathbb{E}_{p(\mathbf{x},\mathbf{z},u)}[\log q_\Omega(\mathbf{x}|\mathbf{z},u)] + \mathbb{E}_{p(\mathbf{x},\mathbf{z},u)}[KL(p(\cdot|\mathbf{z},u)||q_\Omega(\cdot|\mathbf{z},u))] \\
&\geq H(\mathbf{x}) + \mathbb{E}_{p(\mathbf{x},\mathbf{z},u)}[\log q_\Omega(\mathbf{x}|\mathbf{z},u)],
\end{aligned} \tag{14}$$

where $q_\Omega$ is an *arbitrary* auxiliary posterior distribution that aims to maintain the information $\mathbf{x}$ in the representation $\mathbf{z}$ conditioned on the private $u$.

Since $H(\mathbf{x})$ is a constant, our **Goal 2** can be rewritten as the below max-max objective function:

$$\max_f I(\mathbf{x};\mathbf{z}|u) \iff \max_{f,\Omega} \mathbb{E}_{p(\mathbf{x},\mathbf{z},u)}\left[\log q_\Omega(\mathbf{x}|\mathbf{z},u)\right]. \tag{15}$$

*Remark.* We note that Equation (15) can be interpreted as a *cooperative game* between the representation learner $f$ and $q_\Omega$ who aim to preserve the utility collaboratively.

**Maximizing the worst-case MI in Equation (9) for adversarial robustness.** To solve Equation (9), one needs to first find the perturbed data $\mathbf{x}' \in \mathcal{B}_p(\mathbf{x},\epsilon)$ that minimizes MI $I(\mathbf{x}';\mathbf{z}'|u)$, and then maximizes this MI by training the representation learner $f$. As claimed in Zhu et al. (2020); Zhou et al. (2022), minimizing the MI on the worst-case perturbed data is computational challenging. An approximate solution (Zhou et al., 2022) is first performing a strong white-box attack, e.g., the projected gradient descent (PGD) attack (Madry et al., 2018), to generate a set of adversarial examples, and then selecting the adversarial example that has the smallest MI. Assume the strongest adversarial example is $\mathbf{x}^a = \arg\min_{\mathbf{x}' \in \mathcal{B}_p(\mathbf{x},\epsilon)} I(\mathbf{x}';\mathbf{z}'|u)$. The next step is to maximize the MI $\max_f I(\mathbf{x}^a;\mathbf{z}^a|u)$. Zhu et al. (2020) used the MI Neural Estimation (MINE) method (Belghazi et al., 2018) to estimate this MI. Specifically,

$$I(\mathbf{x}^a;\mathbf{z}^a|u) \geq I_\Lambda(\mathbf{x}^a;\mathbf{z}^a|u) = \mathbb{E}_{p(\mathbf{x}^a,\mathbf{z}^a,u)}[t_\Lambda(\mathbf{x}^a,\mathbf{z}^a,u)] - \log\mathbb{E}_{p(\mathbf{x}^a)p(\mathbf{z}^a)p(u)}[\exp(t_\Lambda(\mathbf{x}^a,\mathbf{z}^a,u))], \tag{16}$$

where $t_\Lambda : \mathcal{X} \times \mathcal{Z} \times \{0,1\} \to \mathbb{R}$ can be any family of neural networks parameterized with $\Lambda$. More details about calculating the MI are referred to Section 4.3.

**Objective function of ARPRL.** By using the above MI bounds, the objective function of ARPRL is as follows:

$$\max_f \left(\alpha \min_\Psi - \mathbb{E}_{p(\mathbf{x},u)}[\log q_\Psi(u|f(\mathbf{x}))] + \beta \max_\Lambda I_\Lambda(\mathbf{x}^a;\mathbf{z}^a|u) + (1-\alpha-\beta)\max_\Omega \mathbb{E}_{p(\mathbf{x},u)}[\log q_\Omega(\mathbf{x}|f(\mathbf{x}),u)]\right). \tag{17}$$

where $\alpha,\beta \in [0,1]$ tradeoff between privacy and utility, and robustness and utility, respectively. That is, a larger/smaller $\alpha$ indicates a stronger/weaker attribute privacy protection and a larger/smaller $\beta$ indicates a stronger/weaker robustness against adversarial perturbations.

## 4.3 IMPLEMENTATION IN PRACTICE VIA TRAINING PARAMETERIZED NEURAL NETWORKS

In practice, Equation (17) is solved via training four neural networks, i.e., the representation learner $f_\Theta$ (parameterized with $\Theta$), privacy-protection network $g_\Psi$ associated with the auxiliary distribution $q_\Psi$, robustness network $t_\Lambda$ associated with the MINE estimator, and utility-preservation network $h_\Omega$ associated with the auxiliary distribution $q_\Omega$, on a set of training data.

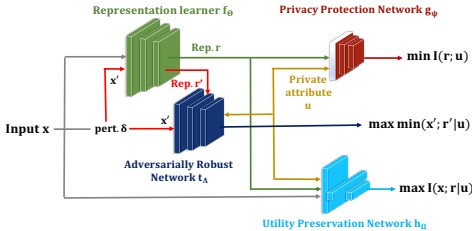

Suppose we have collected a set of samples $\{(\mathbf{x}_j, y_j, u_j)\}$ from the dataset distribution $\mathcal{D}$. We can then approximate each term in Equation (17). Specifically, we approximate the expectation associated with the privacy-protection network network $g_\Psi$ as

Figure 1: Overview of ARPRL.

$$\mathbb{E}_{p(u,\mathbf{x})}\log q_\Psi(u|f(\mathbf{x}))) \approx -\sum_j CE(u_j, g_\Psi(f(\mathbf{x}_j))),$$

where $CE(\cdot)$ means the cross-entropy loss function.

Further, we approximate the expectation associated with the utility-preservation network $h_\Omega$ via the *Jensen-Shannon* (JS) MI estimator (Hjelm et al., 2019). That is,

$$\underset{p(\mathbf{x},u)}{\mathbb{E}} \log q_\Omega(\mathbf{x}|f(\mathbf{x}), u) \approx I^{(JS)}_{\Theta,\Omega}(\mathbf{x}; f(\mathbf{x}), u) = \underset{p(\mathbf{x},u)}{\mathbb{E}} [-\text{sp}(-h_\Omega(\mathbf{x}, f(\mathbf{x}), u)] - \underset{p(\mathbf{x},u,\bar{\mathbf{x}})}{\mathbb{E}} [\text{sp}(h_\Omega(\bar{\mathbf{x}}, f(\mathbf{x}), u)],$$

where $\bar{\mathbf{x}}$ is an independent and random sample from the same distribution as $\mathbf{x}$, and the expectation can be replaced by the samples $\{\mathbf{x}^i_j, \bar{\mathbf{x}}^i_j, u^i_j\}$. $\text{sp}(z) = \log(1 + \exp(z))$ is the softplus function.

Regarding the MI related to the robustness network $t_\Lambda$, we can adopt the methods proposed in Zhu et al. (2020); Zhou et al. (2022). For instance, Zhu et al. (2020) proposed to avoid searching the whole ball, and restrict the search space to be the set of empirical distributions with, e.g., $m$ samples: $\mathcal{S}_m(\epsilon) = \{\frac{1}{m}\sum_{i=1}^m \delta_{\mathbf{x}'_i} : \mathbf{x}'_i \in \mathcal{B}_p(\mathbf{x}_i, \epsilon), \forall i \in [m]\}$. Then it estimates the MI $\min_{\mathbf{x}' \in \mathcal{B}_p(\mathbf{x},\epsilon)} I(\mathbf{x}'; f(\mathbf{x}')|u)$ as

$$\min_{\mathbf{x}'} I^{(m)}_\Lambda(\mathbf{x}'; f(\mathbf{x}')|u) \text{ s.t. } \mathbf{x}' \in \mathcal{S}_m(\epsilon), \tag{18}$$

where $I^{(m)}_\Lambda(\mathbf{x}'; f(\mathbf{x}')|u) = \frac{1}{m}\sum_{i=1}^m t_\Lambda(\mathbf{x}_i, f(\mathbf{x}_i), u_i) - \log[\frac{1}{m}\sum_{i=1}^m e^{t_\Lambda(\bar{\mathbf{x}}_i, f(\mathbf{x}_i), u_i)}]$, where $\{\bar{\mathbf{x}}_i\}$ are independent and random samples that have the same distribution as $\{\mathbf{x}_i\}$.

Zhu et al. (2020) propose an alternating minimization algorithm to solve Equation (18). Specifically, it alternatively performs gradient ascent on $\Lambda$ to maximize $I^{(m)}_\Lambda(\mathbf{x}'; f(\mathbf{x}')|u)$ given $\mathcal{S}_m(\epsilon)$, and then searches for the set of worst-case perturbations on $\{\mathbf{x}'_i : i \in [m]\}$ given $\Lambda$ based on, e.g., projected gradient descent. More details of solving Equation (18) are referred to Zhu et al. (2020).

Figure 1 overviews our ARPRL. Algorithm 1 in Appendix details the training of ARPRL.

## 4.4 THEORETICAL RESULTS

We mainly consider binary private attributes and binary classification. We will leave it as future work to generalize our results to multi-value attributes and multi-class classification.[2] All proofs are in Appendix A.

**Robustness vs. Representation Vulnerability.** We first show the relationship between adversarial risk (or robustness) and representation vulnerability in ARPRL.

**Theorem 2.** *Let all binary task classifiers be $\mathcal{C} = \{C : \mathcal{Z} \to \mathcal{Y}\}$. Then for any representation learner $f : \mathcal{X} \to \mathcal{Z}$, we have*

$$\inf_{C \in \mathcal{C}} \text{AdvRisk}_\epsilon(C \circ f) \geq \frac{1}{\log 2}\big(\text{RV}_\epsilon(f|u) - I(\mathbf{x}; \mathbf{z}|u)\big). \tag{19}$$

*Remark.* Similar to Theorem 1, Theorem 2 shows a smaller representation vulnerability $\text{RV}_\epsilon(f|u)$ indicates a smaller adversarial risk, which means better robustness. In addition, a larger MI $I(\mathbf{x}; \mathbf{z}|u)$ (**Goal 2** for utility preservation) produces a smaller adversarial risk, also implying better robustness.

**Utility vs. Privacy Tradeoff.** The following theorem shows the tradeoff between utility and privacy:

**Theorem 3.** *Let $\mathbf{z} = f(\mathbf{x})$ be with a bounded norm $R$ (i.e., $\max_{\mathbf{z} \in \mathcal{Z}} \|\mathbf{z}\| \leq R$), and $\mathcal{A}$ be the set of all binary inference classifiers that take $\mathbf{z}$ as an input. Assume the task classifier $C$ is $C_L$-Lipschitz, i.e., $\|C\|_L \leq C_L$. Then, we have the below relationship between the standard risk and the advantage:*

$$\text{Risk}(C \circ f) \geq \Delta_{y|u} - 2R \cdot C_L \cdot Adv_\mathcal{D}(\mathcal{A}), \tag{20}$$

*where $\Delta_{y|u} = |Pr_\mathcal{D}(y = 1|u = 0) - Pr_\mathcal{D}(y = 1|u = 1)|$ is a dataset-dependent constant.*

*Remark.* Theorem 3 says that any task classifier using learnt representations incurs a risk on at least a private attribute value. Specifically, the smaller the advantage $\text{Adv}_\mathcal{D}(\mathcal{A})$ (meaning less attribute privacy is leaked), the larger the lower bound risk, and vice versa. Note that the lower bound is independent of the adversary, meaning it covers the *worst-case* attribute inference adversary. Hence, Equation (20) reflects an inherent tradeoff between utility preservation and attribute privacy leakage.

**Robustness vs. Privacy Tradeoff.** Let $\mathcal{D}'$ be a joint distribution over the adversarially perturbed input $\mathbf{x}'$, sensitive attribute $u$, and label $y$. By assuming the representation space is bounded by $R$, the perturbed representations also satisfy $\max_{\mathbf{z}' \in \mathcal{Z}} \|\mathbf{z}'\| \leq R$, where $\mathbf{z}' = f(\mathbf{x}')$. Following Equation 5, we have an associated adversary *advantage* $\text{Adv}_{\mathcal{D}'}(\mathcal{A})$ with respect to the joint distribution $\mathcal{D}'$. Similarly, $\text{Adv}_{\mathcal{D}'}(\mathcal{A}) = 1$ means an adversary can *completely* infer the privacy attribute $u$ through the learnt adversarially perturbed representations $\mathbf{z}'$, and $\text{Adv}_{\mathcal{D}'}(\mathcal{A}) = 0$ implies an adversary only obtains a *random guessing* inference performance. Then we have the following theorem:

---

[2]Zhao et al. (2020) that also has theoretical results of privacy protection against attribute inference attacks. The differences between theirs and our theoretical results are discussed in Appendix A.4.

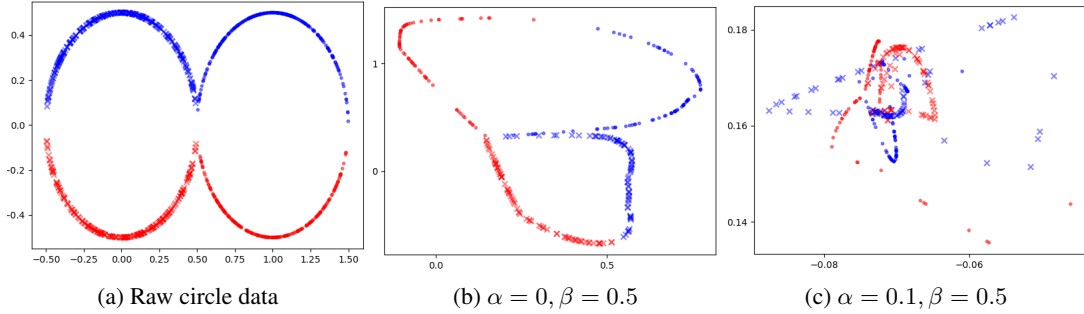

| (a) Raw circle data | (b) $\alpha = 0, \beta = 0.5$ | (c) $\alpha = 0.1, \beta = 0.5$ |

Figure 2: 2D representations learnt by ARPRL. (a) Raw data; (b) only robust representations (privacy acc: 99%, robust acc: 88%, test acc: 99%); and (c) robust + privacy preserving representations (privacy acc: 55%, robust acc: 75%, test acc: 85%). red vs. blue: binary private attribute values; cross $\times$ vs. circle $\circ$: binary task labels.

**Theorem 4.** *Let* $\mathbf{z}' = f(\mathbf{x}')$ *be the learnt representation for* $\mathbf{x}' \in \mathcal{B}(\mathbf{x}, \epsilon)$ *with a bounded norm* $R$ *(i.e.,* $\max_{\mathbf{z}' \in \mathcal{Z}} \|\mathbf{z}'\| \le R$*), and* $\mathcal{A}$ *be the set of all binary inference classifiers. Under a* $C_L$*-Lipschitz task classifier* $C$, *we have the below relationship between the adversarial risk and the advantage:*

$$\text{AdvRisk}_\epsilon(C \circ f) \ge \Delta_{y|u} - 2R \cdot C_L \cdot Adv_{\mathcal{D}'}(\mathcal{A}). \qquad (21)$$

*Remark.* Likewise, Theorem 4 states that, any task classifier using adversarially learnt representations has to incur an adversarial risk on at least a private attribute value. Moreover, the lower bound covers the *worst-case* adversary. Equation (21) hence reflects an inherent tradeoff between adversarial robustness and privacy.

**Guaranteed Attribute Privacy Leakage.** The attribute inference accuracy induced by the worst-case adversary is bounded in the following theorem:

**Theorem 5.** *Let* $\mathbf{z}$ *be the learnt representation by Equation (17). For any attribute inference adversary* $\mathcal{A} = \{A : \mathcal{Z} \to \mathcal{U} = \{0, 1\}\}$, $Pr(A(\mathbf{z}) = u) \le 1 - \frac{H(u|\mathbf{z})}{2\log_2(6/H(u|\mathbf{z}))}$.

*Remark.* Theorem 5 shows that when the conditional entropy $H(u|\mathbf{z})$ is larger, the inference accuracy induced by any adversary is smaller, i.e., less attribute privacy leakage. From another perspective, as $H(u|\mathbf{z}) = H(u) - I(u; \mathbf{z})$, achieving the largest $H(u|\mathbf{z})$ implies minimizing $I(u; \mathbf{z})$ (note that $H(u)$ is a constant)—This is exactly our **Goal 1** aims to achieve.

## 5 EVALUATIONS

We evaluate ARPRL on both synthetic and real-world datasets. The results on the synthetic dataset is for visualization and verifying the tradeoff purpose.

### 5.1 EXPERIMENTAL SETUP

We train the neural networks via Stochastic Gradient Descent (SGD), where the local batch size is 100 and we use 10 local epochs and 50 global epochs in all datasets. The learning rate in SGD is set to be $1e^{-3}$. The detailed network architecture is shown in Table 2 in Appendix B.2. The hyperparameters used in the adversarially robust network are following Zhu et al. (2020). We also discuss how to choose the hyperparameters $\alpha$ and $\beta$ in real-world datasets in Appendix B.3. Without loss of generality, we consider the most challenging $l_\infty$ perturbation. Following Zhu et al. (2020), we use the PGD attack (Madry et al., 2018) for both generating adversarial perturbations in the estimation of worst-case MI and evaluating model robustness[3]. We implement ARPRL in PyTorch and use the NSF Chameleon Cloud GPUs (Keahey et al., 2020) (CentOS7-CUDA 11 with Nvidia Rtx 6000) to train the model. We evaluate ARPRL on three metrics: utility preservation, adversarial robustness, and privacy protection. Our source code will be publicly available upon paper acceptance.

### 5.2 RESULTS ON A TOY EXAMPLE

We generate 2 2D circles with the center (0.0, 0.0) and (1.0, 0.0) respectively, and the radius 0.25, and data points are on the circumference. Each circle indicates a class and has 5,000 samples, where 80% of the samples are for training and the remaining 20% for testing. We define the binary private attribute value for each data point as whether the $y$-value is above or below the x-axis. The network architectures are shown in Table B.2 in Appendix. We use an $l_\infty$ perturbation budget $\epsilon = 0.01$ and 10 PGD attack steps with step size 0.1. We visualize the learnt

---

[3]Note that our goal in this paper is not to design the best adversarial attack, i.e., generating the optimal adversarial perturbation. Hence, the achieved adversarial robustness might not the optimal. We also test CelebA against the CW attack (Carlini & Wagner, 2017), and the robust accuracy is 85%, which close to 87% with the PGD attack.

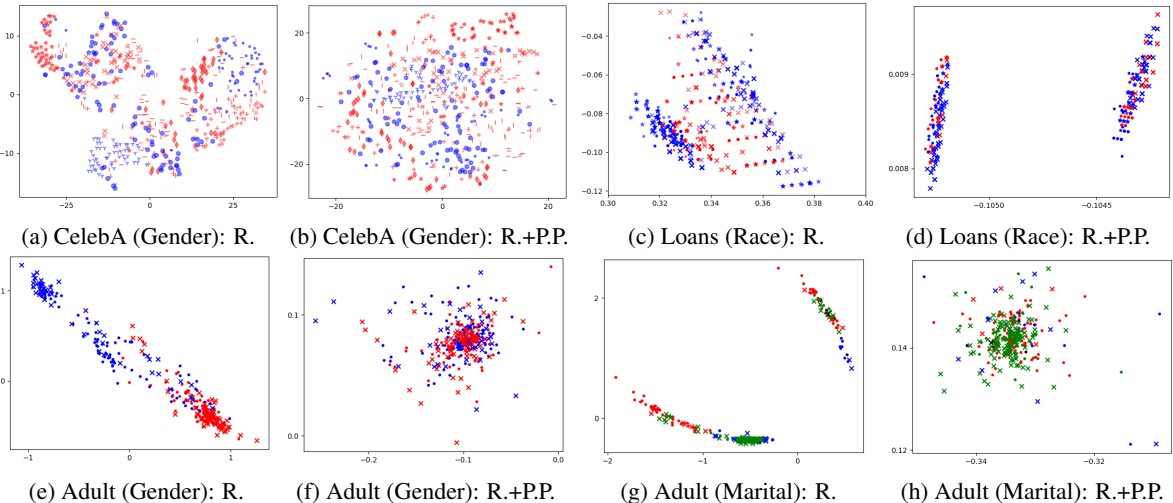

Figure 3: 2D t-SNE representations learnt by ARPRL. *Left:* only learning robust representations; *Right:* learning robust + privacy preserving representations (under the best tradeoff in Table 1). Colors indicate attribute values, while point patterns mean labels.

representations via 2D t-SNE (Van der Maaten & Hinton, 2008) in Figure 2. We can see that: by learning *only robust* representations, the 2-class data can be well separated, but their private attribute values can be also completely separated–almost $100\%$ privacy leakage. In contrast, by learning both *robust and privacy-preserving* representations, the 2-class data can be separated, but their private attributes are mixed—only $55\%$ inference accuracy. Note that the optimal random guessing inference accuracy is $50\%$. We also notice a tradeoff among robustness/utility and attribute privacy, as demonstrated in our theorems. That is, a more robust/accurate model leaks more attribute privacy, and vice versa.

### 5.3 RESULTS ON THE REAL-WORLD DATASETS

**Datasets and setup.** We use three real-world datasets from different applications, i.e., the widely-used CelebA (Liu et al., 2015) image dataset (150K training images and 50K for testing) to study attribute privacy protection (Li et al., 2021), the Loans (Hardt et al., 2016), and Adult Income (Dua & Graff, 2017) datasets. For the CelebA dataset, we treat binary 'gender' as the private attribute, and detect 'gray hair' as the primary (binary classification) task, following Li et al. (2021); Osia et al. (2020b). For the Loans dataset, the primary task is to accurately predict the affordability of the person asking for the loan while protecting their race. Finally, for the Adult Income dataset, predicting whether the income of a person is above $50,000 or not is the primary task. The private attributes are the gender and the marital status. For $l_\infty$ perturbations, we set the budget $\epsilon = 0.01$ for Loans and Adults, and 0.1 for CelebA. We use 10 PGD attack steps with step size 0.1.

**Results.** Tables 1 shows the results on the three datasets, where we report the robust accuracy (under the $l_\infty$ attack), normal test accuracy, and attribute inference accuracy (as well as the gap to random guessing). We have the following observations: 1) When $\alpha = 0$, it means ARPRL only focuses on learning robust representation (similar to (Zhu et al., 2020)) and obtains the best robust accuracy. However, the inference accuracy is rather high, indicating a serious privacy leakage. 2) Increasing $\alpha$ can progressively better protect the attribute privacy, i.e., the inference accuracy is gradually reduced and finally close to random guessing (note different datasets have different random guessing value). 3) $\alpha$ and $\beta$ together act as the tradeoff among robustness, utility, and privacy. Particularly, a better privacy protection (i.e., larger $\alpha$) implies a smaller test accuracy, indicating an utility and privacy tradeoff, as validated in Theorem 3. Similarly, a better privacy protection also implies a smaller robust accuracy, indicating a robustness and privacy tradeoff, as validated in Theorem 4.

**Visualization.** We further visualize the learnt representations via t-SNE in Figure 3. We can see that: When only focusing on learning robust representations, both the data with different labels and with different attribute values can be well separated. On the other hand, when learning both robust and privacy-preserving representations, the data with different labels can be separately, but they are mixed in term of the attribute values—meaning the privacy of attribute values is protected to some extent.

**Runtime.** We only show runtime on the largest CelebA (150K training images). In our used platform, it took about 5 mins each epoch (about 15 hours in total) to learn the robust and privacy-preserving representation for each hyperparameter setting. The computational bottleneck is mainly from training robust representations (where we adapt the source code from Zhu et al. (2020)), which occupies 60% of the training time (e.g., 3 mins out of 5 mins in each epoch). Training the other neural networks is much faster.

Table 1: Test accuracy, robust accuracy, vs. inference accuracy (and gap w.r.t. the optimal random guessing) on the considered three datasets and private attributes. Note that some datasets are unbalanced, so the random guessing values are different. Larger $\alpha$ means more privacy protection, while larger $\beta$ means more robustness against adversarial perturbation. $\alpha = 0$ means no privacy protection and only focuses on robust representation learning, same as (Zhu et al., 2020; Zhou et al., 2022).

| CelebA | | | | |
|---|---|---|---|---|
| Private attr.: Gender (binary), budget $\epsilon = 0.1$ | | | | |
| $\alpha$ | $\beta$ | Rob. Acc | Test Acc | Infer. Acc (gap) |
| 0 | 0.50 | 0.87 | 0.91 | 0.81 (0.31) |
| 0.1 | 0.45 | 0.84 | 0.88 | 0.75 (0.25) |
| 0.5 | 0.25 | 0.79 | 0.85 | 0.62 (0.12) |
| 0.9 | 0.05 | 0.71 | 0.81 | 0.57 (0.07) |

| Loans | | | | |
|---|---|---|---|---|
| Private attr.: Race (binary), budget $\epsilon = 0.01$ | | | | |
| $\alpha$ | $\beta$ | Rob. Acc | Test Acc | Infer. Acc (gap) |
| 0 | 0.50 | 0.45 | 0.74 | 0.92 (0.22) |
| 0.05 | 0.475 | 0.42 | 0.69 | 0.75 (0.05) |
| 0.10 | 0.45 | 0.40 | 0.68 | 0.72 (0.02) |
| 0.15 | 0.425 | 0.39 | 0.66 | 0.71 (0.01) |

| Adult income | | | | |
|---|---|---|---|---|
| $\alpha$ | $\beta$ | Rob. Acc | Test Acc | Infer. Acc (gap) |
| Private attr.: Gender (binary), budget $\epsilon = 0.01$ | | | | |
| 0 | 0.5 | 0.63 | 0.68 | 0.88 (0.33) |
| 0.05 | 0.475 | 0.57 | 0.67 | 0.72 (0.17) |
| 0.10 | 0.45 | 0.55 | 0.65 | 0.59 (0.04) |
| 0.20 | 0.4 | 0.53 | 0.63 | 0.55 (0.00) |

| Adult income | | | | |
|---|---|---|---|---|
| $\alpha$ | $\beta$ | Rob. Acc | Test Acc | Infer. Acc (gap) |
| Private attr.: Marital status (7 values), budget $\epsilon = 0.01$ | | | | |
| 0 | 0.5 | 0.56 | 0.71 | 0.70 (0.14) |
| 0.001 | 0.495 | 0.55 | 0.65 | 0.60 (0.04) |
| 0.005 | 0.49 | 0.52 | 0.60 | 0.59 (0.03) |
| 0.01 | 0.45 | 0.47 | 0.59 | 0.57 (0.01) |

## 5.4 COMPARING WITH THE STATE-OF-THE-ARTS

**Comparing with task-known privacy-protection baselines.** We compare ARPRL with two recent task-known methods for attribute privacy protection on CelebA: **DPFE** (Osia et al., 2020b) that also uses mutual information (but in different ways) and **Deepobfuscator** (Li et al., 2021) that is adversarial training based defense. Specifically, we ensure the three methods have the same test accuracy 0.88, and compare the attribute inference accuracy. *For fair comparison, we do not consider adversarial robustness in our ARPRL.* The attribute inference accuracy of DPFE and Deepobfuscator are 0.79 and 0.70, respectively, and our ARPRL's is 0.71. First, DPFE performs much worse because it assumes the distribution of the learnt representation to be Gaussian (which could be inaccurate), while Deepobfuscator and ARPRL do not have any assumption on the distributions; Second, Deepobfuscator performs slightly better than ARPRL. This is because both ARPRL and Deepobfuscator involve adversarial training, Deepobfuscator uses task labels, but ARPRL is task-agnostic, hence slightly sacrificing privacy.

**Comparing with task-known adversarial robustness baselines.** We compare ARPRL with the state-of-the-art task-known adversarial training based TRADES (Zhang et al., 2019) and test on CelebA, under the same adversarial perturbation and without privacy-protection (i.e., $\alpha = 0$). For task-agnostic ARPRL, its robust accuracy is 0.87, which is slightly worse than TRADES's is 0.89. However, when ARPRL also includes task labels during training, its robust accuracy increases to 0.91—This again verifies that adversarially robust representations based defenses outperform the classic adversarial training based method.

**Comparing with task-known TRADES + Deepobfuscator for both robustness and privacy protection.** A natural solution to achieve both robustness and privacy protection is by combining the SOTAs that are individually adversarially robust or privacy-preserving. Here, we test TRADES + Deepopfuscator on CelebA. By tuning the tradeoff hyperparameters, we obtain the best utility, privacy, and robustness tradeoff of TRADES + Deepopfuscator as: (Robust Acc, Test Acc, Infer. Acc) = (0.79, 0.84, 0.65). In contrast, the best tradeoff of ARPRL in Table 1 is (Robust Acc, Test Acc, Inference Acc) = (0.79, 0.85, 0.62), which is slightly better than TRADES + Deepopfuscator, though they both know the task labels. The results imply that simply combining SOTA robust and privacy-preserving methods is not the best option. Instead, our ARPRL learns both robust and privacy-preserving representations under the same information-theoretic framework.

## 6 CONCLUSION AND FUTURE WORK

In this paper, we aim to ensure machine learning models to be robust against adversarial examples and protect sensitive attributes in the data. We achieve the goal by proposing ARPRL, which learns adversarially robust, privacy preserving, and utility preservation representations under a unified information-theoretic framework. We also derive theoretical results that show the inherent tradeoff between robustness/utility and privacy and guarantees of attribute privacy against the worst-case attribute inference adversary. ARPRL is also shown to outperform the state-of-the-arts via empirical evaluations.

Future works include 1) generalizing the results to other well-known security attacks such as data poisoning attack and backdoor attack, and other well-known privacy attacks such as membership inference attack and data reconstruction attack; 2) evaluating ARPRL on other data modalities such as audio, speech, and natural language; 3) generalizing theoretical results to multi-value attributes and provable robustness guarantees.

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

---

**Algorithm 1** Adversarially robust and privacy-preserving representation learning (**ARPRL**)

---

**Input:** A dataset $\mathcal{D} = \{\mathbf{x}_i, y_i, u_i\}$, perturbation budget $\epsilon$, $\alpha, \beta \in [0, 1]$, $\lambda > 0$, learning rates $lr_1, lr_2, lr_3, lr_4, lr_5$; #global epochs $I$, #local gradient steps $J$;
**Output:** Representation learner parameters $\Theta$.

1: Initialize $\Theta, \Psi, \Omega, \Lambda$ for the representation learner $f_\Theta$, privacy protection network $g_\Phi$, utility preservation network $h_\Omega$, and adversarially robust network $t_\Lambda$;
2: **for** $t = 0; t < T; t + +$ **do**
3:     $L_1 = \sum_i CE(u_i, g_\Psi(f(\mathbf{x}_i)))$;
4:     $L_2 = \frac{1}{|\mathcal{D}|} \sum_i t_\Lambda(\mathbf{x}_i, f_\Theta(\mathbf{x}_i), u_i) - \log[\frac{1}{|\mathcal{D}|} \sum_i e^{t_\Lambda(\bar{\mathbf{x}}_i, f_\Theta(\mathbf{x}_i), u_i)}]$;
5:     $L_3 = I_{\Theta,\Omega}^{(JS)}(\mathbf{x}; f(\mathbf{x}), u)$;
6:     **for** $i = 0; i < I; i + +$ **do**
7:         **for** $j = 0; j < J; j + +$ **do**
8:             $\Psi \leftarrow \Psi - lr_1 \cdot \frac{\partial L_1}{\partial \Psi}$;
9:             $\Lambda \leftarrow \Lambda + lr_2 \cdot \frac{\partial L_2}{\partial \Lambda}$;
10:            $\Omega \leftarrow \Omega + lr_3 \cdot \frac{\partial L_3}{\partial \Omega}$;
11:         **end for**
12:         $\Theta \leftarrow \Theta + lr_4 \cdot \frac{\partial(\alpha L_1 + \beta L_2 + (1-\alpha-\beta)L_3)}{\partial \Theta}$;
13:     **end for**
14: **end for**
    =0

---

# A PROOFS

## A.1 PROOF OF THEOREM 2

**Theorem 2.** *Let all binary task classifiers be* $\mathcal{C} = \{C : \mathcal{Z} \to \mathcal{Y}\}$. *Then for any representation learner* $f : \mathcal{X} \to \mathcal{Z}$, *we have*

$$\inf_{C \in \mathcal{C}} \text{AdvRisk}_\epsilon(C \circ f) \geq \frac{1}{\log 2} \big(\text{RV}_\epsilon(f|u) - I(\mathbf{x}; \mathbf{z}|u)\big). \tag{19}$$

*Proof.* Replacing $I(\mathbf{x}; \mathbf{z})$ and $\text{RV}_\epsilon(f)$ in Theorem 1 with $I(\mathbf{x}; \mathbf{z}|u)$ and $\text{RV}_\epsilon(f|u)$, and setting $|\mathcal{Y}| = 2$, we reach Theorem 2. $\square$

## A.2 PROOF OF THEOREM 3

We first introduce the following definitions and lemmas that will be used to prove Theorem 3.

**Definition 1** (Lipschitz function and Lipschitz norm). *We say a function* $f : A \to \mathbb{R}^m$ *is L-Lipschitz continuous, if for any* $a, b \in A$, $\|f(a) - f(b)\| \leq L \cdot \|a - b\|$. *Lipschitz norm of* $f$, *i.e.,* $\|f\|_L$, *is defined as* $\|f\|_L = \max \frac{\|f(a) - f(b)\|_L}{\|a - b\|_L}$.

**Definition 2** (Total variance (TV) distance). *Let* $\mathcal{D}_1$ *and* $\mathcal{D}_2$ *be two distributions over the same sample space* $\Gamma$, *the TV distance between* $\mathcal{D}_1$ *and* $\mathcal{D}_2$ *is defined as:* $d_{TV}(\mathcal{D}_1, \mathcal{D}_2) = \max_{E \subseteq \Gamma} |\mathcal{D}_1(E) - \mathcal{D}_2(E)|$.

**Definition 3** (1-Wasserstein distance). *Let* $\mathcal{D}_1$ *and* $\mathcal{D}_2$ *be two distributions over the same sample space* $\Gamma$, *the 1-Wasserstein distance between* $\mathcal{D}_1$ *and* $\mathcal{D}_2$ *is defined as* $W_1(\mathcal{D}_1, \mathcal{D}_2) = \max_{\|f\|_L \leq 1} |\int_\Gamma f d\mathcal{D}_1 - \int_\Gamma f d\mathcal{D}_2|$, *where* $\|\cdot\|_L$ *is the Lipschitz norm of a real-valued function.*

**Definition 4** (Pushforward distribution). *Let* $\mathcal{D}$ *be a distribution over a sample space and* $g$ *be a function of the same space. Then we call* $g(\mathcal{D})$ *the pushforward distribution of* $\mathcal{D}$.

**Lemma 1** (Contraction of the 1-Wasserstein distance). *Let* $g$ *be a function defined on a space and* $L$ *be constant such that* $\|g\|_L \leq C_L$. *For any distributions* $\mathcal{D}_1$ *and* $\mathcal{D}_2$ *over this space,* $W_1(g(\mathcal{D}_1), g(\mathcal{D}_2)) \leq C_L \cdot W_1(\mathcal{D}_1, \mathcal{D}_2)$.

**Lemma 2** (1-Wasserstein distance on Bernoulli random variables). *Let* $y_1$ *and* $y_2$ *be two Bernoulli random variables with distributions* $\mathcal{D}_1$ *and* $\mathcal{D}_2$, *respectively. Then,* $W_1(\mathcal{D}_1, \mathcal{D}_2) = |Pr(y_1 = 1) - Pr(y_2 = 1)|$.

**Lemma 3** (Relationship between the 1-Wasserstein distance and the TV distance (Gibbs & Su, 2002)). *Let* $g$ *be a function defined on a norm-bounded space* $\mathcal{Z}$, *where* $\max_{\mathbf{z} \in \mathcal{Z}} \|\mathbf{z}\| \leq R$, *and* $\mathcal{D}_1$ *and* $\mathcal{D}_1$ *are two distributions over the space* $\mathcal{Z}$. *Then* $W_1(g(\mathcal{D}_1), g(\mathcal{D}_2)) \leq 2R \cdot d_{TV}(g(\mathcal{D}_1), g(\mathcal{D}_2))$.

**Lemma 4** (Relationship between the TV distance and advantage Liao et al. (2021)). *Given a binary attribute* $u \in \{0, 1\}$. *Let* $\mathcal{D}_{u=a}$ *be the conditional data distribution of* $\mathcal{D}$ *given* $u = a$ *over a sample space* $\Gamma$. *Let* $Adv_\mathcal{D}(\mathcal{A})$ *be the advantage of adversary. Then for any function* $f$, *we have* $d_{TV}(f(\mathcal{D}_{u=0}), f(\mathcal{D}_{u=1})) = Adv_\mathcal{D}(\mathcal{A})$.

We now prove Theorem 3, which is restated as below:

**Theorem 3.** *Let $\mathbf{z} = f(\mathbf{x})$ be with a bounded norm $R$ (i.e., $\max_{\mathbf{z} \in \mathcal{Z}} \|\mathbf{z}\| \leq R$), and $\mathcal{A}$ be the set of all binary inference classifiers that take $\mathbf{z}$ as an input. Assume the task classifier $C$ is $C_L$-Lipschitz, i.e., $\|C\|_L \leq C_L$. Then, we have the below relationship between the standard risk and the advantage:*

$$\text{Risk}(C \circ f) \geq \Delta_{y|u} - 2R \cdot C_L \cdot Adv_{\mathcal{D}}(\mathcal{A}), \tag{20}$$

*where $\Delta_{y|u} = |Pr_{\mathcal{D}}(y = 1|u = 0) - Pr_{\mathcal{D}}(y = 1|u = 1)|$ is a dataset-dependent constant.*

*Proof.* We denote $\mathcal{D}_{u=a}$ as the conditional data distribution of $\mathcal{D}$ given $u = a$, and $\mathcal{D}_{y|u}$ as the conditional distribution of label $y$ given $u$. $cf$ is denoted as the (binary) composition function of $c \circ f_\Theta$. As $c$ is binary task classifier on the learnt representations, it follows that the pushforward $cf(\mathcal{D}_{u=0})$ and $cf(\mathcal{D}_{u=1})$ induce two distributions over the binary label space $\mathcal{Y} = \{0, 1\}$. By leveraging the triangle inequalities of the 1-Wasserstein distance, we have

$$
\begin{aligned}
&W_1(\mathcal{D}_{y|u=0}, \mathcal{D}_{y|u=1}) \\
&\leq W_1(\mathcal{D}_{y|u=0}, cf(\mathcal{D}_{u=0})) + W_1(cf(\mathcal{D}_{u=0}), cf(\mathcal{D}_{u=1})) + W_1(cf(\mathcal{D}_{u=1}), \mathcal{D}_{y|u=1})
\end{aligned}
\tag{22}
$$

Using Lemma 2 on Bernoulli random variables $y|u = a$:

$$W_1(\mathcal{D}_{y|u=0}, \mathcal{D}_{y|u=1}) = |\text{Pr}_{\mathcal{D}}(y = 1|u = 0) - \text{Pr}_{\mathcal{D}}(y = 1|u = 1)| = \Delta_{y|u}. \tag{23}$$

Using Lemma 1 on the contraction of the 1-Wasserstein distance and that $\|c\|_L \leq C_L$, we have

$$W_1(cf(\mathcal{D}_{u=0}), cf(\mathcal{D}_{u=1})) \leq C_L \cdot W_1(f(\mathcal{D}_{u=0}), f(\mathcal{D}_{u=1})). \tag{24}$$

Using Lemma 3 with $\max_{\mathbf{z}} \|\mathbf{z}\| \leq R$, we have

$$W_1(f(\mathcal{D}_{u=0}), f(\mathcal{D}_{u=1})) \leq 2R \cdot d_{TV}(f(\mathcal{D}_{u=0}), f(\mathcal{D}_{u=1})) = 2R \cdot \text{Adv}_{\mathcal{D}}(\mathcal{A}), \tag{25}$$

where the last equation is based on Lemma 4.

Combing Equations 24 and 25, we have

$$W_1(cf(\mathcal{D}_{u=0}), cf(\mathcal{D}_{u=1})) \leq 2R \cdot C_L \cdot \text{Adv}_{\mathcal{D}}(\mathcal{A}).$$

Furthermore, using Lemma 2 on Bernoulli random variables $y$ and $cf(\mathbf{x})$, we have

$$
\begin{aligned}
&W_1(\mathcal{D}_{y|u=a}, cf(\mathcal{D}_{u=a})) \\
&= |\text{Pr}_{\mathcal{D}}(y = 1|u = a) - \text{Pr}_{\mathcal{D}}(cf(\mathbf{x}) = 1|u = a))| \\
&= |\mathbb{E}_{\mathcal{D}}[y|u = a] - \mathbb{E}_{\mathcal{D}}[cf(\mathbf{x})|u = a]| \\
&\leq \mathbb{E}_{\mathcal{D}}[|y - cf(\mathbf{x})||u = a] \\
&= \text{Pr}_{\mathcal{D}}(y \neq cf(\mathbf{x})|u = a) \\
&= \text{Risk}_{u=a}(c \circ f).
\end{aligned}
\tag{26}
$$

Hence, $W_1(\mathcal{D}_{y|u=0}, cf(\mathcal{D}_{u=0})) + W_1(\mathcal{D}_{y|u=1}, cf(\mathcal{D}_{u=1})) \leq \text{Risk}(c \circ f)$.

Finally, by combining Equations (22) - (26), we have:

$$\Delta_{y|u} \leq \text{Risk}(c \circ f) + 2R \cdot C_L \cdot \text{Adv}_{\mathcal{D}}(\mathcal{A}), \tag{27}$$

thus $\text{Risk}(c \circ f) \geq \Delta_{y|u} - 2R \cdot C_L \cdot \text{Adv}_{\mathcal{D}}(\mathcal{A})$, completing the proof.

$\square$

### A.3 PROOF OF THEOREM 4

We follow the way as proving Theorem 3. We first restate Theorem 4 as below:

**Theorem 4.** *Let $\mathbf{z}' = f(\mathbf{x}')$ be the learnt representation for $\mathbf{x}' \in \mathcal{B}(\mathbf{x}, \epsilon)$ with a bounded norm $R$ (i.e., $\max_{\mathbf{z}' \in \mathcal{Z}} \|\mathbf{z}'\| \leq R$), and $\mathcal{A}$ be the set of all binary inference classifiers. Under a $C_L$-Lipschitz task classifier $C$, we have the below relationship between the adversarial risk and the advantage:*

$$\text{AdvRisk}_\epsilon(C \circ f) \geq \Delta_{y|u} - 2R \cdot C_L \cdot Adv_{\mathcal{D}'}(\mathcal{A}). \tag{21}$$

*Proof.* Recall that $D'$ is a joint distribution of the perturbed input $\mathbf{x}'$, the label $y$, and private attribute $u$. We denote $\mathcal{D}'_{u=a}$ as the conditional perturbed data distribution of $\mathcal{D}'$ given $u = a$, and $\mathcal{D}'_{y|u}$ as the conditional distribution of label $y$ given $u$. Also, the pushforward $cf(\mathcal{D}'_{u=a})$ induces two distributions over the binary label space $\mathcal{Y} = \{0, 1\}$ with $a = \{0, 1\}$. Via the triangle inequalities of the 1-Wasserstein distance, we have

$$W_1(\mathcal{D}'_{y|u=0}, \mathcal{D}'_{y|u=1})$$
$$\leq W_1(\mathcal{D}'_{y|u=0}, cf(\mathcal{D}'_{u=0})) + W_1(cf(\mathcal{D}'_{u=0}), cf(\mathcal{D}'_{u=1})) + W_1(cf(\mathcal{D}'_{u=1}), \mathcal{D}'_{y|u=1}) \tag{28}$$

Using Lemma 2 on Bernoulli random variables $y|u = a$:

$$W_1(\mathcal{D}'_{y|u=0}, \mathcal{D}'_{y|u=1}) = |\text{Pr}_{\mathcal{D}'}(y = 1|u = 0) - \text{Pr}_{\mathcal{D}'}(y = 1|u = 1)|$$
$$= |\text{Pr}_{\mathcal{D}}(y = 1|u = 0) - \text{Pr}_{\mathcal{D}}(y = 1|u = 1)| = \Delta_{y|u}, \tag{29}$$

where we use that the perturbed data and clean data share the same label $y$ condition on $u$.

Then following the proof of Theorem 3, we have:

$$W_1(cf(\mathcal{D}'_{u=0}), cf(\mathcal{D}'_{u=1})) \leq C_L \cdot W_1(f(\mathcal{D}'_{u=0}), f(\mathcal{D}'_{u=1})); \tag{30}$$
$$W_1(f(\mathcal{D}'_{u=0}), f(\mathcal{D}'_{u=1})) \leq 2R \cdot d_{TV}(f(\mathcal{D}'_{u=0}), f(\mathcal{D}'_{u=1})). \tag{31}$$

We further show $d_{TV}(f(\mathcal{D}'_{u=0}), f(\mathcal{D}'_{u=1})) = \text{Adv}_{\mathcal{D}'}(\mathcal{A})$:

$$d_{TV}(f(\mathcal{D}'_{u=0}), f(\mathcal{D}'_{u=1}))$$
$$= \max_E |\text{Pr}_{f(\mathcal{D}'_{u=0})}(E) - \text{Pr}_{f(\mathcal{D}'_{u=1})}(E)|$$
$$= \max_{A_E \in \mathcal{A}} |\text{Pr}_{\mathbf{z}' \sim f(\mathcal{D}'_{u=0})}(A_E(\mathbf{z}') = 1) - \text{Pr}_{\mathbf{z}' \sim f(\mathcal{D}'_{u=1})}(A_E(\mathbf{z}) = 1)|$$
$$= \max_{A_E \in \mathcal{A}} |\text{Pr}(A_E(\mathbf{z}') = 1|u = 0) - \text{Pr}(A_E(\mathbf{z}') = 1|u = 1)|$$
$$= \text{Adv}_{\mathcal{D}'}(\mathcal{A}), \tag{32}$$

where the first equation uses the definition of TV distance, and $A_E(\cdot)$ is the characteristic function of the event $E$ in the second equation.

With Equations (30) - (32), we have

$$W_1(cf(\mathcal{D}'_{u=0}), cf(\mathcal{D}'_{u=1})) \leq 2R \cdot C_L \cdot \text{Adv}_{\mathcal{D}'}(\mathcal{A}).$$

Furthermore, using Lemma 2 on Bernoulli random variables $y$ and $cf(\mathbf{x})$, we have

$$W_1(\mathcal{D}'_{y|u=0}, cf(\mathcal{D}'_{u=0})) + W_1(\mathcal{D}'_{y|u=1}, cf(\mathcal{D}'_{u=1}))$$
$$= |\text{Pr}_{\mathcal{D}'}(y = 1|u = 0) - \text{Pr}_{\mathcal{D}'}(cf(\mathbf{x}') = 1|u = 0))| + |\text{Pr}_{\mathcal{D}'}(y = 1|u = 1) - \text{Pr}_{\mathcal{D}'}(cf(\mathbf{x}') = 1|u = 1))|$$
$$= |\mathbb{E}_{\mathcal{D}'}[y|u = 0] - \mathbb{E}_{\mathcal{D}'}[cf(\mathbf{x}')|u = 0]| + |\mathbb{E}_{\mathcal{D}'}[y|u = 1] - \mathbb{E}_{\mathcal{D}'}[cf(\mathbf{x}')|u = 1]|$$
$$\leq \mathbb{E}_{\mathcal{D}'}[|y - cf(\mathbf{x}')||u = 0] + \mathbb{E}_{\mathcal{D}'}[|y - cf(\mathbf{x}')||u = 1]$$
$$= \text{Pr}_{\mathcal{D}'}(y \neq cf(\mathbf{x}')|u = 0) + \text{Pr}_{\mathcal{D}'}(y \neq cf(\mathbf{x}')|u = 1)$$
$$= \text{Pr}_{\mathcal{D}'}(y \neq cf(\mathbf{x}'))$$
$$= \text{Pr}_{\mathcal{D}}[\exists \mathbf{x}' \in \mathcal{B}(\mathbf{x}, \epsilon), \text{ s.t. } cf(\mathbf{x}') \neq y]$$
$$= \text{AdvRisk}_\epsilon(c \circ f). \tag{33}$$

Finally, by combining Equations (28) - (33), we have:

$$\Delta_{y|u} \leq \text{AdvRisk}_\epsilon(c \circ f) + 2R \cdot C_L \cdot \text{Adv}_{\mathcal{D}'}(\mathcal{A})$$

Hence, $\text{AdvRisk}_\epsilon(c \circ f) \geq \Delta_{y|u} - 2R \cdot C_L \cdot \text{Adv}_{\mathcal{D}'}(\mathcal{A})$, completing the proof.

$\square$

## A.4 PROOF OF THEOREM 5

We first point out that Zhao et al. (2020) also provide the theoretical result in Theorem 3.1 against attribute inference attacks. However, there are two key differences between theirs and our Theorem 5: First, Theorem 3.1 requires an assumption $I(\hat{A}; A|Z) = 0$, while our Theorem 5 does not need extra assumption; 2) The proof for Theorem 3.1 decomposes an joint entropy $H(A, \hat{A}, E)$, while our proof decomposes a conditional entropy $H(s, u|A(z))$. We note that the main idea to prove both theorems is by introducing an indicator and decomposing an entropy in two different ways.

The following lemma about the inverse binary entropy will be useful in the proof of Theorem 5:

**Lemma 5** ((Calabro, 2009) Theorem 2.2). *Let $H_2^{-1}(p)$ be the inverse binary entropy function for $p \in [0, 1]$, then $H_2^{-1}(p) \geq \frac{p}{2 \log_2(\frac{6}{p})}$.*

**Lemma 6** (Data processing inequality). *Given random variables $X$, $Y$, and $Z$ that form a Markov chain in the order $X \to Y \to Z$, then the mutual information between $X$ and $Y$ is greater than or equal to the mutual information between $X$ and $Z$. That is $I(X; Y) \geq I(X; Z)$.*

With the above lemma, we are ready to prove Theorem 5 restated as below.

**Theorem 5.** *Let $\mathbf{z}$ be the learnt representation by Equation (17). For any attribute inference adversary $\mathcal{A} = \{A : \mathcal{Z} \to \mathcal{U} = \{0, 1\}\}$, $Pr(A(\mathbf{z}) = u) \leq 1 - \frac{H(u|\mathbf{z})}{2 \log_2(6/H(u|\mathbf{z}))}$.*

*Proof.* Let $s$ be an indicator that takes value 1 if and only if $\mathcal{A}(\mathbf{z}) \neq u$, and 0 otherwise, i.e., $s = 1[\mathcal{A}(\mathbf{z}) \neq u]$. Now consider the conditional entropy $H(s, u|\mathcal{A}(\mathbf{z}))$ associated with $\mathcal{A}(\mathbf{z})$, $u$, and $s$. By decomposing it in two different ways, we have

$$H(s, u|\mathcal{A}(\mathbf{z})) = H(u|\mathcal{A}(\mathbf{z})) + H(s|u, \mathcal{A}(\mathbf{z})) = H(s|\mathcal{A}(\mathbf{z})) + H(u|s, \mathcal{A}(\mathbf{z})). \tag{34}$$

Note that $H(s|u, \mathcal{A}(\mathbf{z})) = 0$ as when $u$ and $\mathcal{A}(\mathbf{z})$ are known, $s$ is also known. Similarly,

$$\begin{aligned} H(u|s, \mathcal{A}(\mathbf{z})) &= Pr(s = 1)H(u|s = 1, \mathcal{A}(\mathbf{z})) + Pr(s = 0)H(u|s = 0, \mathcal{A}(\mathbf{z})) \\ &= 0 + 0 = 0, \end{aligned} \tag{35}$$

because when we know $s$'s value and $\mathcal{A}(\mathbf{z})$, we also actually knows $u$.

Thus, Equation 34 reduces to $H(u|\mathcal{A}(\mathbf{z})) = H(s|\mathcal{A}(\mathbf{z}))$. As conditioning does not increase entropy, i.e., $H(s|\mathcal{A}(\mathbf{z})) \leq H(s)$, we further have

$$H(u|\mathcal{A}(\mathbf{z})) \leq H(s). \tag{36}$$

On the other hand, using mutual information and entropy properties, we have $I(u; \mathcal{A}(\mathbf{z})) = H(u) - H(u|\mathcal{A}(\mathbf{z}))$ and $I(u; \mathbf{z}) = H(u) - H(u|\mathbf{z})$. Hence,

$$I(u; \mathcal{A}(\mathbf{z})) + H(u|\mathcal{A}(\mathbf{z})) = I(u; \mathbf{z}) + H(u|\mathbf{z}). \tag{37}$$

Notice $\mathcal{A}(\mathbf{z})$ is a random variable such that $u \perp \mathcal{A}(\mathbf{z})|\mathbf{z}$. Hence, we have the Markov chain $u \to \mathbf{z} \to \mathcal{A}(\mathbf{z})$. Based on the data processing inequality in Lemma 6, we know $I(u; \mathcal{A}(\mathbf{z})) \leq I(u; \mathbf{z})$. Combining it with Equation 37, we have

$$H(u|\mathcal{A}(\mathbf{z})) \geq H(u|\mathbf{z}). \tag{38}$$

Combing Equations (36) and (38), we have $H(s) = H_2(Pr(s = 1)) \geq H(u|\mathbf{z})$, which implies

$$Pr(\mathcal{A}(\mathbf{z}) \neq u) = Pr(s = 1) \geq H_2^{-1}(H(u|\mathbf{z})),$$

where $H_2(t) = -t \log_2 t - (1 - t) \log_2(1 - t)$.

Finally, by applying Lemma 5, we have

$$Pr(\mathcal{A}(\mathbf{z}) \neq u) \geq \frac{H(u|\mathbf{z})}{2 \log_2(6/H(u|\mathbf{z}))}.$$

Hence the attribute privacy leakage is bounded by $Pr(\mathcal{A}(\mathbf{z}) = u) \leq 1 - \frac{H(u|\mathbf{z})}{2 \log_2(6/H(u|\mathbf{z}))}$.

$\square$

## B   DATASETS AND NETWORK ARCHITECTURES

### B.1   DETAILED DATASET DESCRIPTIONS

**CelebA dataset (Liu et al., 2015).** CelebA consists of more than 200K face images with size 32x32. Each face image is labeled with 40 binary facial attributes. In the experiments, we use 150K images for training and 50K images for testing. We treat binary 'gender' as the private attribute, and detect 'gray hair' as the primary (binary classification) task.

**Loans dataset (Hardt et al., 2016).** This dataset is originally extracted from the loan-level Public Use Databases. The Federal Housing Finance Agency publishes these databases yearly, containing information about the

Table 2: Network architectures for the used datasets. Note that utility preservation network is the same as robust network.

| Representation Learner | Robust Network | Privacy Network | Utility Network |
|---|---|---|---|
| CelebA | | | |
| conv1-64 | conv3-64 | linear-32 MaxPool | conv3-64 |
| conv64-128 | conv64-128 | linear-#priv. attri. values | conv64-128 |
| linear-1024 | conv128-256 | | conv128-256 |
| linear-64 | conv2048-2048 | | conv2048-2048 |
| Loans and Adult Income | | | |
| linear-12 | linear-64 | linear-16 | linear-64 |
| linear-3 | linear-3 | linear-#priv. attri. values | linear-3 |
| Toy dataset | | | |
| linear-10 | linear-64 | linear-5 | linear-64 |
| linear-2 | linear-2 | linear-#priv. attri. values | linear-2 |

Enterprises' single family and multifamily mortgage acquisitions. Specifically, the database used in this project is a single-family dataset and has a variety of features related to the person asking for a mortgage loan. All the attributes in the dataset are numerical, so no preprocessing from this side was required. On the other hand, in order to create a balanced classification problem, some of the features were modified to have a similar number of observations belonging to all classes. We use 80% data for training and 20% for testing.

The utility under this scope was measured in the system accurately predicting the affordability category of the person asking for a loan. This attribute is named *Affordability*, and has three possible values: 0 if the person belongs to a mid-income family and asking for a loan in a low-income area, 1 if the person belongs to a low-income family and asking for a loan in a low-income area, and 2 if the person belongs to a low-income family and is asking for a loan not in a low-income area. The private attribute was set to be binary *Race*, being White (0) or Not White (1).

**Adult Income dataset (Dua & Graff, 2017).** This is a well-known dataset available in the UCI Machine Learning Repository. The dataset contains 32,561 observations each with 15 features, some of them numerical, other strings. Those attributes are not numerical were converted into categorical using an encoder. Again, we use the 80%-20% train-test split.

The primary classification task is predicting if a person has an income above $50,000, labeled as 1, or below, which is labeled as 0. The private attributes to predict are the *Gender*, which is binary, and the *Marital Status*, which has seven possible labels: 0 if Divorced, 1 if AF-spouse, 2 if Civil-spouse, 3 if Spouse absent, 4 if Never married, 5 if Separated, and 6 if Widowed.

### B.2 NETWORK ARCHITECTURES

The used network architectures for the three neural networks are in Table 2.

### B.3 HOW TO CHOOSE $\alpha$ AND $\beta$

Assume we reach the required utility with a (relatively large) value $1 - \alpha - \beta$ (e.g., 0.7, 0.8; note its regularization controls the utility). Then we have a principled way to efficiently tune $\alpha$ and $\beta$ based on their meanings:

1) We will start with three sets of $(\alpha 1, \beta 1), (\alpha 2, \beta 2), (\alpha 3, \beta 3)$, where one is with $\alpha 1 = \beta 1$, one is with a larger $\alpha 2 > \alpha 1$ (i.e., better privacy), and one is with a larger $\beta 3 > \beta 1$ (better robustness), respectively.

2) Based on the three results, we know whether a larger $\alpha$ or $\beta$ is needed to obtain a better privacy-robustness tradeoff and set their values via a binary search. For instance, if needing more privacy protection, we can set a larger $\alpha 4 = \frac{\alpha 1 + \alpha 2}{2}$; or needing more robustness, we can set a larger $\beta 4 = \frac{\beta 1 + \beta 3}{2}$.

Step 2) continues until finding the optimal tradeoff $\alpha$ and $\beta$.

