# OpenReview forum: "Adversarially Robust and Privacy-Preserving Representation Learning via Information Theory"
_ICLR.cc/2024/Conference — ICLR 2024 Conference Withdrawn Submission_

### Official Review · Reviewer_qq4Y · 2023-10-31

**Soundness:** 2 fair
**Presentation:** 2 fair
**Contribution:** 1 poor
**Rating:** 3
**Confidence:** 4

**Summary:**

The authors propose a method that is both adversarially robust as well as good at private attribute obfuscation while ensuring the information with respect to the utility variable is intact. They claim that unlike prior works they merge the two objectives using information theory.

**Strengths:**

the overall idea of the paper is quite relevant as we need to unify the notion of robustness to adversaries of all sorts in a training procedure. However several questions remain unanswered in the current work. Hope the review comments here help authors improve their current manuscript.

**Weaknesses:**

- It is not clear from the paper how the different networks and losses of each network are implemented. E.g., Fig. 1 still defines networks in terms of mutual information while the networks themselves are actually approximating those via cross-entropy and MINE estimator. It would be better to also show the corresponding losses the networks are optimizing over.
- Just like the adversarial private representation learning, the proposed method ARPRL does not have privacy guarantees.
- Also while the authors start the initial analysis using an information theoretic approach they eventually resort to upper bounds that result in various cross-entropy losses and a MINE estimator. These upper and lower bounds are known to be quite loose. So in effect, while the authors claim that this is driven by information theory, the final optimization seems to be quite disjoint from this and depends on several networks some of which are adversarial while others are cooperative.
- Some of the results are quite similar to prior works:
	- information theoretic analysis is quite similar to [1]
	- The intuition about random guessing based on eqn 5 is quite similar to [2]
- Some possible use cases/baselines that exist but have not been considered by the authors include
	- Also unlike some prior works such as [2] the proposed method can only work with one utility and one sensitive target.
	- closed-form solutions by [3] give very robust benchmarks to compare against
- The proposed method is also not compared against private learning algorithms if the representation learner is trained with adversarial examples. I believe a simple augmentation to existing works such as [1, 2, 3] might serve as good baselines to compare against.

1. Azam, Sheikh Shams, et al. "Can we generalize and distribute private representation learning?." _International Conference on Artificial Intelligence and Statistics_. PMLR, 2022.
2. Bertran, Martin, et al. "Adversarially learned representations for information obfuscation and inference." _International Conference on Machine Learning_. PMLR, 2019.
3. Sadeghi, Bashir, and Vishnu Naresh Boddeti. "Imparting fairness to pre-trained biased representations." _Proceedings of the IEEE/CVF Conference on Computer Vision and Pattern Recognition Workshops_. 2020.

**Questions:**

Please see the weaknesses above.

---

> ### Author Response · Authors · 2023-11-20
> **Response to the Reviewer**
>
> We thank the reviewer for the constructive comments!
>
>
> **Comment#1**: How different networks and losses of each network are implemented.
>
> **Response**: The neural network losses are defined in Section 4.3, and the detailed training procedure is in Algorithm 1 in Appendix.
>
>
> **Comment#2**: Just like the adversarial private representation learning, the proposed method ARPRL does not have privacy guarantees.
>
> **Response**: This is a misunderstanding! **Theorem 5** shows the information-theoretic privacy guarantee.
>
>
> **Comment#3**: mutual information upper and lower bounds are known to be quite loose; the final optimization seems to be quite disjoint from mutual information goals.
>
> **Response**: We admit that the estimated mutual information bounds are loose. The fundamental reason lies in the challenges of accurately calculating the mutual information for high-dimensional random variables. We emphasize the state-of-the-art mutual information neural estimators (e.g., Cheng et al. 2020) also mention the gap, but they show the (loose) mutual information bounds do not affect the performance too much in practice.
>
> **Comment#4**: information theoretic analysis is quite similar to [1] Azam et al. AISTATS 2022; The intuition about random guessing based on eqn 5 is quite similar to [2]
>
>
> **Response**: We carefully read the paper [1], but do not see any information theoretical results. Could you please have a double check?
>
>
> We put the advantage definition in Eqn 5 in the preliminary section and hence do not treat it as our contribution. We also point out that the advantage definition is widely used in the literature.
>
>
> **Comment#5**: the proposed method can only work with one utility and one sensitive target
>
> **Response:** In the paper, we only focus on a sensitive attribute and a primary task, which is most widely studied.. However, our method can be easily generalized to multiple sensitive attributes. For instance, when there is a set of $B$ private attributes $\{u_1, u_2, \cdots, u_B \}$ to be protected, our Goal 1 can be extended to $\min_f \sum_{b=1}^B I(z;u_b)$, while Goal 2 can be extended as $\max_f I(x;z|u_1, u_2, \cdots u_b)$.
>
> **Comment#6:** closed-form solutions by [3] (CVPR workshop 2020) give very robust benchmarks to compare against
>
> **Response:** The closed-form solution in [3] is obtained by assuming the representation learner to be *linear* to the input, i.e., $z = \theta x$, which limits the learning capability. In
> Our compared methods are more recent (e.g., Deepobfuscator (Li et al., 2021)), which shows state-of-the-art privacy protection performance.
>
> **Comment#7:** The proposed method is also not compared against private learning algorithms if the representation learner is trained with adversarial examples.
>
> **Response:** As suggested, we choose [2] as the private learner and augment adversarial examples with the PGD attack. We test [2] + adversarial examples on CelebA. By tuning the tradeoff hyperparameters, we obtain the best utility, privacy, and robustness tradeoff of [2] + adversarial examples as: (Robust Acc, Test Acc, Infer. Acc) = (0.73, 0.82, 0.62). In contrast, the best tradeoff of ARPRL in is (Robust Acc, Test Acc, Inference Acc) = (0.79, 0.85, 0.62). Still, our results show simply combining adversarial examples and privacy-preserving methods is not the best option.

---

> > ### Comment · Reviewer_qq4Y · 2023-11-23
> > **Response to Authors**
> >
> > Thank you for the response. Below is my response to author clarifications.
> >
> > > The neural network losses are defined in Section 4.3, and the detailed training procedure is in Algorithm 1 in Appendix.
> >
> > I agree with the authors that the training information is indeed available in the paper, but is however hard to follow. Also, it would be important to move Algorithm 1 early into the paper, given that the overall training process is not as simple as running a forward-backward pass through a single network.
> >
> > The main point I wish to put across is that the current presentation of the paper makes it really hard to follow the paper, especially section 4. For example,
> > - figure 1 mentions that the network $g_{\psi}$ needs to minimize $I(r;u)$ where $r = f_{\theta}(x)$ but in section 4 when explaining the design, authors refer to it as $z=f_{\theta}(x)$
> > - I believe there is a typo when authors write $\max \min (x^\prime; r^\prime\vert u)$ in the figure for network $t_{\Lambda}$
> > - eqn (17) has multiple $\min$ and $\max$ without parentheses affecting the readability. Also, what happens if the $\alpha + \beta > 1$?
> >
> > >  Theorem 5 shows the information-theoretic privacy guarantee
> >
> > While theorem 5 gives information-theoretic guarantees, these information-theoretic metrics are not evaluated in closed form but approximated by their optimization counterparts, e.g., CE loss and MINE estimator. Do the authors mean that Theorem 5 applies directly to the networks that are trained? I believe it would be dependent on the gaps of these bounds that are estimated by these networks as well as the convergence of each network during training?
> >
> > Another idea that the authors might consider is that the representation network uses adversarial perturbation and it might be possible for authors to also get differential privacy guarantees.
> >
> > > We admit that the ... affect the performance too much in practice.
> >
> > I agree with the authors about the gap in the bound. However, I believe it becomes more important to analyze these gaps given the nature of current work being related to privacy.
> >
> > > We carefully read the paper [1], but do not see any information theoretical results. Could you please have a double check?
> >
> > I would like to apologize for the incorrect citation indices, it should be " information theoretic analysis is quite similar to [2] Bertran et al. ICML 2019; The intuition about random guessing based on eqn 5 is quite similar to [1] Azam et al. AISTATS 2022".
> >
> > Please check the section 2.1 in [2], i.e., Bertran, Martin, et al. "Adversarially learned representations for information obfuscation and inference." International Conference on Machine Learning. PMLR, 2019. Pointing towards the lemmas 2.1, 2.2, and 2.3.
> >
> > > We put the advantage definition ... widely used in the literature.
> >
> > Adding citations to the literature might be helpful for the readers. Also, citations to works with similar intuition would further bolster the insights that the authors present here.
> >
> > > ... However, our method can be easily generalized to multiple sensitive attributes. ...
> >
> > Do the authors think that such generalization will impact the architecture presented in Fig.1 and in turn the optimization? I'd assume the addition of more terms to goals 1 and 2 would further require a careful understanding of its implication on the bounds that are derived in section 4 for the practical optimization of the networks. Furthermore, the effect of such a generalization on the MINE estimator is also not easy to extrapolate from existing results in literature.
> >
> > > ... our results show simply combining adversarial examples and privacy-preserving methods is not the best option.
> >
> > That is indeed a very interesting result. And I thank the authors for undertaking the experiment to answer my question. Another question that arises is with regards to how private these representations are with respect to a stronger unseen adversary. For example, consider the attack experiment similar to Table 2 of Azam et al. AISTATS 2022.
> >
> > Also, in line with the response of the other reviewer, I agree that most of the existing techniques in this domain have several hyperparameters that would lead to a trade-off curve, thus making it an unfair comparison to consider only one checkpoint of hyperparameter values.
> >
> > In my opinion, while the current paper has strengths, several key questions remain open with the manuscript in its current form. So, I'd keep my current score unchanged.

---

### Official Review · Reviewer_jNV8 · 2023-11-15

**Soundness:** 2 fair
**Presentation:** 2 fair
**Contribution:** 2 fair
**Rating:** 3
**Confidence:** 3

**Summary:**

The paper studied the setting of combining adversarially attack and privacy-preserving problems together and tackled this from an information theoretic perspective. Specifically, the author is inspired by the setting of adversarial perturbation under the framework of representation vulnerability and defines the notion of attribute inference advantage. The paper aims for three different goals, which are utility preservation, privacy, and adversarially robustness, and achieves them by formulating the objectives using mutual information, it further proposes the model ARPRL, by using different types of neural-network-based mutual information estimators. The paper studied the tradeoffs between utility, adversarial robustness, and privacy, by drawing several theoretical conclusions. Finally, the paper evaluated the method’s performance and the tradeoff between utility, adversarial robustness, and privacy on synthetic and real-world datasets.

**Strengths:**

The main strength of the paper is the formulation of studying both adversarial robustness and privacy and the systematic study of these tradeoffs through both theoretical analysis and experiments.

**Weaknesses:**

While the paper's formulation of the problem and the systematic study are of importance, the current version has several weaknesses.

1. The presentation can be generally further improved. Several places in the current paper need further clarification, the details can be found in the Questions section.

2. While the theoretical analysis is of importance, currently it is limited to the binary setting, which can be not practical.

3. The paper writing can be further improved. Firstly, the paper studied several different objectives, including utility, adversarial robustness, and privacy, but there is not a consistent order of introduction in different sections, for example, in Sections 2 and 3, the order is first adversarial robustness then privacy. While in Section 4, the order becomes first privacy, then utility, then adversarial robustness. Also, both adversarial and privacy objectives are defined as something with an abbreviation "Adv". One stands for adversarial, and the other stands for advantage. These can introduce difficulty in understanding while reading.

**Questions:**

**Method:**

I'm having difficulty understanding the reason for conditioning on u for formalizing Goal 2 and Goal 3, as the first goal's objective already minimizes the I(z,u), why does the method need to maximize/minimize the conditional mutual information instead of just the mutual information? Also, for example, considering Goal 3 alone, should we want the representation z to be robust to the adversarial perturbance regarding all the mutual information I(x;z) instead of just the mutual information excluding u while excluding u could make the representation not truly adversarially robust? What is the benefit of conditioning on u for formalizing Goal 2 and Goal 3?

Besides, if not conditioning on u, then the combination of Goal 1 and Goal 2, will also be closely related to an information bottleneck objective, of which its relation to privacy (privacy-utility tradeoff) has been discussed in some previous literature, such as [1].

This also raises the question of the necessity to use four different neural networks to achieve the goal of this paper.  The training of neural network-based mutual information method can require interactive training between MI estimation and maximization/minimization and training several different mutual information-based objectives at the same time could result in unnecessary unstability.

Should Equation 14 (first line as an example) be I(x;z|u) = H(x|u) - H(x|z,u)?


**Experiments:**

In Section 5.4, the first paragraph "Comparing with task-known privacy-protection baselines", what is the alpha and beta used in this setting? The result with test acc 0.88 and infer acc 0.71 is not shown in the table. Is the result of ARPRL Section 5.4 computed as the optimal result from different combinations of alpha and beta? How sensitive is the result to different values of alpha and beta?

For the second paragraph "Comparing with task-known adversarial robustness baselines", as the proposed method is described as a task-agnostic method in the previous sections, how does the method include task labels during training in this setting?

For the third paragraph, what is the definition of the best "utility, privacy, and robustness tradeoff"? Why is it just one set of results?


Besides, it says in the paper that the whole network is trained using SGD using learning rate 1e−3. Does it include all four networks (which means in Algorithm 1 lr1=lr2=lr3=lr4) or does each network need some extra different finetuning? As there are four different networks in the model, it would be good to discuss how sensitive is the training concerning these hyperparameters.

Minor problem: I think the lr5 in Algorithm 1 is defined but not used.

[1] Makhdoumi, Ali, et al. "From the information bottleneck to the privacy funnel." 2014 IEEE Information Theory Workshop (ITW 2014). IEEE, 2014.

---

> ### Author Response · Authors · 2023-11-20
> **Response to the Reviewer**
>
> We thank the reviewer for appreciating the theoretical contributions and constructive comments!
>
> **Comment#1**: Why conditioning on u for formalizing Goal 2 and Goal 3, as Goal 1's objective already minimizes the I(z,u). Excluding u could make the representation not truly adversarially robust.
>
> **Response**:  **Conditioning on $u$ for formulating Goal 2**: The MI in Goal 1 is to reduce the correlation between the private $u$ and representation $z$. The purpose of conditioning on $u$ is to also remove the information that $x$ contains about the private $u$, not just only letting $z$ be uncorrelated to $u$ in an uncontrollable way. Hene, together with Goal 1, $z$ can maintain more information in $x$, as well as leaking less information in $u$. Without such a condition on $u$, the formulation is the same as the privacy tunnel (Makhdoumi, et al. [1]). Note that the compared DPFE (Osia et al., 2020b) indeed uses the idea as privacy tunnel [1] to learn privacy-preserving representations. Our results show DPFE performs worse than our ARPRL against privacy protection, which demonstrates that the conditioning on $u$ is necessary.
>
> **Conditioning on $u$ for formulating Goal 3**: The purpose is to ensure that the learnt representation $z$ is robust to adversarial perturbation, *under the constraint that the private $u$ cannot be leaked from $z$*. Hence, this can produce a representation that is both robust and privacy-preserving.
>
>
> **Comment#2:** Should Equation 14 (first line as an example) be I(x;z|u) = H(x|u) - H(x|z,u)?
>
> **Response**: Yes, it is! Thanks for your correction!
>
> **Comment#3:**  "Comparing with task-known privacy-protection baselines", what is the $\alpha$ and $\beta$?
>
> **Response:** $\alpha=0.1$ and $\beta=0$, as we only consider privacy protection (in order to fairly compare with the privacy-protection baselines). That is also why infer acc=0.71 is not shown in Table 1, where $\alpha=0.1$ and $\beta=0.45$.
>
> **Comment#4:** Is the result of ARPRL Section 5.4 computed as the optimal result from different combinations of $\alpha$ and $\beta$? How sensitive is the result to different values of $\alpha$ and $\beta$?
>
> **Response:** No, the results in Section 5.4 are irrelevant to the optimal results. They are obtained by directly training the neural networks with the corresponding $\alpha$ and $\beta$.
>
> The experimental results in Table 1 show that $\alpha$ controls privacy, while $\beta$ controls robustness. Our findings were that, given a fixed $\alpha$ (or $\beta$), the results are relatively insensitive to a small range of $\beta$ (or $\alpha$).
>
> **Comment#5:** Comparing with task-known adversarial robustness baselines, how does the proposed method include task labels during training.
>
> **Response:** We emphasize that our ARPRL does NOT use task labels during training, since it is task-agnostic.
>
> **Comment#6:** What is the definition of the best "utility, privacy, and robustness tradeoff"? Why is it just one set of results?
>
> **Response:** Our theoretical results show that utility, privacy, and robustness have inherent tradeoffs. Empirically, a better tradeoff means a larger utility, smaller inference accuracy, and larger robust accuracy at the same time. Note that the tradeoffs can be reflected by tuning $\alpha$ and $\beta$ values (i.e., a larger/smaller $\alpha$ indicates a stronger/weaker attribute privacy protection and a larger/smaller $\beta$ indicates a stronger/weaker robustness against adversarial perturbations). We do not say (and cannot prove theoretically) the best tradeoff is obtained at one single set of  $\alpha$ and $\beta$. What we can do in practice is to tune $\alpha$ and $\beta$ (See B.3 in Appendix) and observe the trend of the tradoff.
>
>
> **Comment#7:**  same learning rate 1e−3 in all neural networks? How sensitive the hyperparameters are.
>
> **Response:** Yes, we use the same learning rate. We admit that it is difficult to tune the four neural networks simultaneously in practice to obtain the *optimal* performance. We also observed that the adversarial robustness network is sensitive to a large learning rate (e.g., when lr=0.01, the robustness performance could drop 10-15\% among the four datasets). To ensure a stable training, we hence consider relatively *small* learning rates in all networks and set an equal value 1e−3 for simplicity. Our results also showed this value exhibits promising performance in terms of utility,  adversarial robustness, and privacy protection.
>
> **Comment#8:** Minors: change the section order, lr5 is not used.
>
> **Response:** Thanks for pointing this out. Will fix them.

---

> > ### Comment · Reviewer_jNV8 · 2023-11-22
> > **Thanks for the response. Follow-up questions.**
> >
> > I thank the authors for their responses and clarifying answers. My follow-ups are as follows:
> >
> > **Comment#1**:
> >
> > ***Conditioning on $u$ for formulating Goal 2***
> >
> > I thank the authors for their clarifying answers to the connection to previous work. A follow-up question is that in Section 5.4, the paper says that DPFE performs badly mainly because of the assumption that the distribution of the learned representation is Gaussian, so the performance gap here does not effectively justify that not conditioning on u will harm the performance.
> >
> > ***Conditioning on $u$ for formulating Goal 3***
> >
> > To my understanding, instead of constraint, it is more accurate to say is "under the assumption that the private $u$ cannot be leaked from $z$ (which may not hold depending on the specific setting and how well the training goes)”, so I still not convinced that conditioning on $u$ is the most suitable objective in Goal 3 here.
> >
> > **Comment#4**:
> >
> > I’m still not sure I fully understand how alpha and beta are chosen here. Take “Comparing with task-known adversarial robustness baselines” in Section 5.4 as an example, the alpha is set to be 0 as we are considering the case without privacy protection, then how do we choose beta to do the comparison here?
> >
> > The author says that the performance is relatively insensitive when alpha or beta is fixed. It would be good if there is a quantitative measure of the sensitivity here.
> >
> > **Comment#5**:
> >
> > Yes, I understand the method is task-agnostic. The question here is referring to the sentence “However, when ARPRL also includes task labels during training, its robust accuracy increases to 0.91” in Section 5.4 of the paper.
> >
> > **Comment#6**:
> >
> > If it is a trend for the tradeoff, then showing one single set of numbers, instead of a spectrum of numbers, will not be enough for demonstrating one method is better than the other.
> >
> >
> > I’ve read the other review and I’m still convinced that the work needs to be further improved. I’m going to keep my current score unchanged.

---

### Official Review · Reviewer_qrh8 · 2023-11-23

**Soundness:** 3 good
**Presentation:** 2 fair
**Contribution:** 2 fair
**Rating:** 5
**Confidence:** 3

**Summary:**

The paper tackles the question of training models that are simultaneously robust to adversarial examples, and to attribute inference attacks. An objective function is proposed based on several (conditional) mutual information terms, one associated with each of three objectives (utility, robustness to adversarial examples, privacy). The objective is optimized using (previously proposed) approximations to these MI terms.
A theoretical analysis shows that there are inherent trade-offs (for any learned representation) between these objectives.

**Strengths:**

- The problem is well-motivated and relevant to the community.
- The objective function is well-motivated, and the authors made an effort to give an intuitive explanation.
- The paper explains how it builds on prior work.
- The synthetic experiments further contribute to the intuition. The empirical evaluation on real data shows promising performance.

**Weaknesses:**

- The overall presentation needs improvement. The paper is centered around three objectives (utility, adversarial robustness, attribute inference) and these seem to appear in arbitrary and inconsistent order throughout the different sections. The paper can benefit from a more careful organization.
- It seems that most (all?) of the results in section 4.4 state bounds that hold for any representation f (under norm constraints etc.), rather than for the particular learned representation that optimizes the proposed objective (17). The statements of Thm 4 and 5 seem to indicate that this is specific to the particular representation optimizing (17), but I don't see evidence of this in the proof. Please clarify.
- The remark following Thm 3 and 4 are unclear. What is meant by "a risk on at least a private attribute value", and (in Thm 4) what does the "adversarially learnt representation" refer to?
- Most of the conclusions offered by Section 4.4 are not surprising. For example, that maximizing the conditional entropy H(u|z) reduces vulnerability to attribute attack seems unsurprising. I expected that the analysis would offer some quantitative characterization of the trade-offs (ideally in terms of the $\alpha, \beta$ parameters in the optimized objective) but this is not the case. In summary, I am unsure what the goal of the analysis is precisely. Is it to provide additional motivation for the proposed objective? (motivation was already provided in Section 3).
- In the experiments, there is a missed opportunity to precisely explore the trade-off between the three objectives. For example in Table 1 alpha and beta are changed simultaneously. Why choose these particular values of alpha, beta?
- In the experiments: there seems to be some amount of speculation. Some conclusions are presented as facts without evidence. For example, it's unclear how the authors attribute the poor performance of DPFE to its assumption that the learned representation is Gaussian. It's also unclear why being task agnostic would necessarily mean sacrificing privacy.
- Minor: (i) there are several typos. (ii) It's unclear why restricting to binary private attributes is without loss of generality.

**Questions:**

- I invite the authors to comment on the interpretation of the results of Section 4.4 and what they bring to the picture (see points above).
- About the attack model that the paper considers: On the one hand, the "robustness objective" seems to assume that the adversary has access to the input x (since the adversary optimizes over a ball centered at x). On the other hand, the "privacy leakage" objective seems to assume that the adversary only has access to z but not x (since the classifier is assumed to only take z as input). This discrepancy needs some discussion. Can you clarify the attack model?